# RARE-TO-FREQUENT: UNLOCKING COMPOSITIONAL GENERATION POWER OF DIFFUSION MODELS ON RARE CONCEPTS WITH LLM GUIDANCE

**Dongmin Park[1], Sebin Kim[2], Taehong Moon[1], Minkyu Kim[1], Kangwook Lee[1,3], Jaewoong Cho[1]**
[1]KRAFTON, [2]Seoul National University, [3]University of Wisconsin-Madison

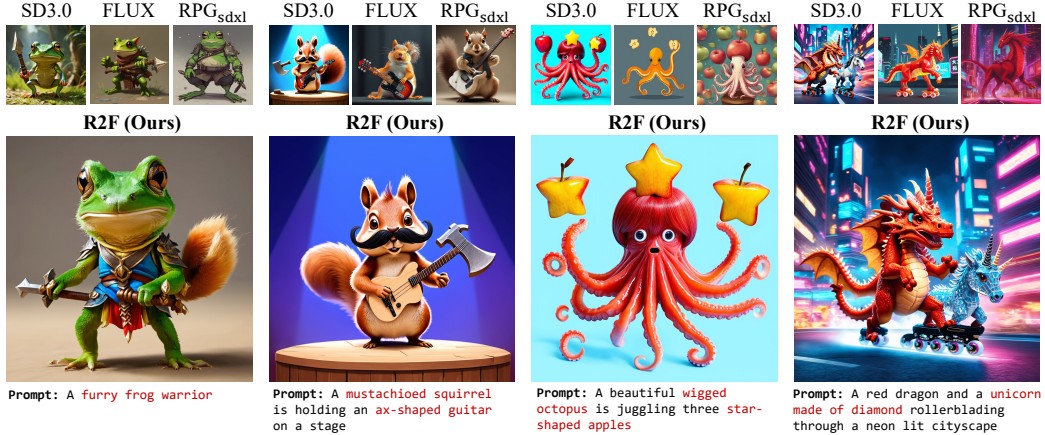

Figure 1: Generated images from prompts with *rare compositions of concepts* (=attribute+object; highlighted in red). These objects possess attributes not typically associated with them, making such combinations difficult to observe. While state-of-the-art pre-trained and LLM-grounded text-to-image diffusion models, SD3.0 (Esser et al., 2024), FLUX (BlackForestLabs, 2024), and RPG (Yang et al., 2024), struggle to generate such concepts, our *training-free* approach, R2F, exhibits superior results.

## ABSTRACT

State-of-the-art text-to-image (T2I) diffusion models often struggle to generate rare compositions of concepts, e.g., objects with unusual attributes. In this paper, we show that the compositional generation power of diffusion models on such rare concepts can be significantly enhanced by the Large Language Model (LLM) guidance. We start with empirical and theoretical analysis, demonstrating that exposing frequent concepts relevant to the target rare concepts during the diffusion sampling process yields more accurate concept composition. Based on this, we propose a training-free approach, R2F, that plans and executes the overall rare-to-frequent concept guidance throughout the diffusion inference by leveraging the abundant semantic knowledge in LLMs. Our framework is flexible across any pre-trained diffusion models and LLMs, and seamlessly integrated with the region-guided diffusion approaches. Extensive experiments on three datasets, including our newly proposed benchmark, RareBench, containing various prompts with rare compositions of concepts, R2F significantly surpasses existing models including SD3.0 and FLUX by up to $28.1\%p$ in T2I alignment. Code is available at link.

## 1 INTRODUCTION

Recent advancements in text-to-image (T2I) diffusion models have achieved unprecedented success in generating highly realistic and diverse images (Zhang et al., 2023a; Saharia et al., 2022). However, these models often struggle to accurately generate images from rare and complex prompts (Samuel et al., 2024; Parashar et al., 2024). Accordingly, many studies have focused on enhancing their *T2I alignment* performance (Rassin et al., 2024). Recently, several approaches have further tried to ground large language models (LLMs) into diffusion models, so-called *LLM-grounded* diffusion models, showing state-of-the-art results by effectively leveraging LLMs' knowledge in diffusion inference.

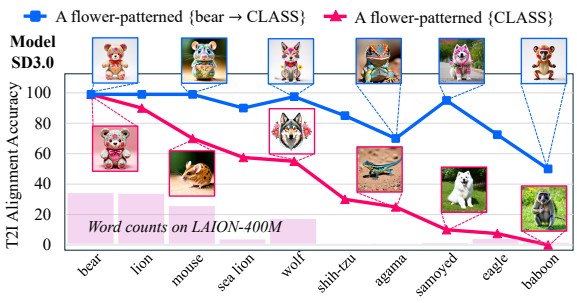
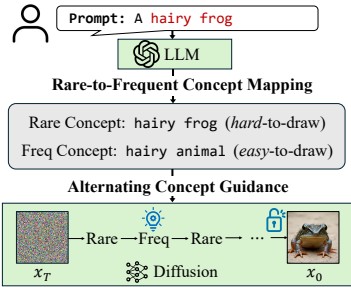

(a) Effectiveness of frequent concept exposure.

(b) Key idea of R2F.

Figure 2: (a) shows image generation quality on a rare composition of two concepts; "*flower-patterned*" and some "*animal*" (randomly sampled from ImageNet classes). Naive inferences with SD3.0 (red line) tend to be inaccurate when the composition becomes rarer (animal classes rarely appear on LAION dataset). Interestingly, once we guide the inference with a relatively frequent composition ("*flower-patterned bear*", which is easily generated as "*bear doll*") at the early sampling steps and then turn back to the original prompt, the generation quality is significantly enhanced (blue line). (b) shows the key idea of our framework with LLM guidance.

In general, LLM-grounded diffusion models use LLMs to decompose a given prompt into sub-prompts for each object and obtain their proper location (e.g., bounding box), and then generate each sub-prompt at its designated location using region-controlled diffusion methods (Yang et al., 2023). While this allows better spatial composition, the models still struggle to generalize on *non-spatially compositional rare* concepts. For example, as in Figure 1, state-of-the-art diffusion models, SD3.0 (Esser et al., 2024), FLUX (BlackForestLabs, 2024) and RPG (Yang et al., 2024), do *not* accurately generate such concepts, such as "*furry* frog warrior" and "*ax-shaped* guitar". However, since real user creators often want to generate such rare concepts (Kirstain et al., 2023), e.g., designing a new cartoon character with creative and unprecedented attributes, this calls for a new approach to further leverage LLMs' knowledge of non-spatial text semantics for image generation.

Our study starts from the following research question: *Do pre-trained diffusion models possess the potential power to compose rare concepts, and can this be unlocked by a training-free approach?*. To answer this, we first explore a controlled experiment for compositional T2I generation. As in Figure 2(a), for a rare composition of two concepts, "*flower-patterned*" and some "*animal*", SD3.0 struggles more and more as the concept composition becomes rarer (with animal classes rarely appear on LAION-400M (Schuhmann et al., 2022)). However, we observe that, by simply exposing a relatively frequent concept composition ("*flower-patterned bear*") at a few early diffusion sampling steps, the model's compositional ability is significantly enhanced. We further provide a theoretical analysis for this phenomenon using the score estimator (Song & Ermon, 2019) in Section 3.1. Therefore, finding appropriate frequent concepts and using them in inference can be a key to enhanced rare concept compositions.

Based on this, we propose a novel approach, called **Rare-to-Frequent (R2F)**, that leverages an LLM to find frequent concepts relevant to rare concepts in prompts and uses them to guide diffusion inference, enabling more precise image synthesis. Specifically, LLM decomposes the given prompt into sub-prompts per object and finds if any rare concepts are in each sub-prompt. If rare concepts are detected, LLM finds their relevant yet frequent alternatives, which are easier to be generated by diffusion models. Then, the diffusion model alternately exposes rare and frequent prompts during the early stages of diffusion, where the LLM also determines the proper stop point based on the visual detail levels required to draw each concept. Note that, R2F is *flexible* to any LLMs and diffusion architectures, and we further propose its seamless integration with region-guided diffusion models, called R2F+, enabling more controlled image generation.

To thoroughly validate the efficacy of R2F, we present a new benchmark, dubbed **RareBench**, consisting of diverse and complex rare compositions of concepts. On RareBench and two existing compositionality benchmarks (Rassin et al., 2024; Huang et al., 2023), R2F outperforms state-of-the-art diffusion baselines, including SD3.0, IterComp, and FLUX, by up to $28.1\%p$ in terms of T2I alignment accuracy. Moreover, we show that R2F can generate images of rare concepts that existing models almost always fail to generate even with careful prompt paraphrasing, showcasing the superiority of our framework in unlocking the compositional generation power of diffusion models.

## 2 RELATED WORK

### 2.1 TEXT-TO-IMAGE DIFFUSION MODELS

Diffusion models are a promising class of generative models and have shown remarkable success in T2I synthesis (Sohl-Dickstein et al., 2015; Ho et al., 2020; Zhang et al., 2023a). Owing to large-scale datasets and pre-trained text embedding models such as CLIP (Radford et al., 2021), GLIDE (Nichol et al., 2021) and Imagen (Saharia et al., 2022) show diffusion models can understand text semantics at scale and synthesize high-quality images. Latent Diffusion Models (LDMs) (Rombach et al., 2022) improve the training efficiency by changing the diffusion process from pixel to latent space. Recently, more advanced models such as SDXL (Podell et al., 2023), PixArt (Chen et al., 2023), and SD3.0 (Esser et al., 2024) further enhance the T2I synthesis quality by using enhanced datasets (Betker et al., 2023), architectures (Peebles & Xie, 2023), and training schemes (Lipman et al., 2022).

### 2.2 COMPOSITIONAL IMAGE GENERATION

Despite decent advances, recent T2I diffusion models often suffer from image compositionality issues (Rassin et al., 2024; Huang et al., 2023). Many approaches have tried to mitigate this issue based on *cross-attention control* technique. Some works utilize prior linguistic knowledge for text token-level attention control. StructureDiffusion (Feng et al., 2022), Attend-and-Excite (Chefer et al., 2023), and SynGen (Rassin et al., 2024) control text tokens of different objects to be located in separate attention regions. Another line of work uses additional input conditions. GLIGEN (Li et al., 2023) and ReCo (Yang et al., 2023) propose position-aware adapters attachable to the diffusion backbone, and use regional conditions to locate each object at the corresponding region. ControlNet (Zhang et al., 2023b) and InstanceDiffusion (Wang et al., 2024b) introduce more general adapters that incorporate diverse conditions. While these works have succeeded in advancing compositional generation, they require prior linguistic knowledge or extra conditions, which are hard to prepare for arbitrary text.

### 2.3 LLM-GROUNDED DIFFUSION

LLMs have shown promising abilities in language comprehension (Achiam et al., 2023; Zhao et al., 2023; Wang et al., 2024a). Powered by this advance, recent works have attempted to ground LLMs into diffusion models to provide prior knowledge and conditions for compositional image generation. LayoutGPT (Feng et al., 2024), LMD (Lian et al., 2023), and CompAgent (Wang et al., 2024d) use LLMs to decompose a given prompt into sub-prompts per object and extract their corresponding bounding boxes. RPG (Yang et al., 2024) further adopts recaptioning and planning for complementary regional diffusion. ELLA (Hu et al., 2024) use LLMs to dynamically extract timestep-dependent conditions from intricate prompts. Some work utilizes LLMs in iterative image editing (Wu et al., 2024; Gani et al., 2023; Wang et al., 2024c). While these LLM-grounded diffusion approaches have succeeded in spatial compositions, they still struggle in *non-spatial* compositions for *rare* concepts.

## 3 RARE-TO-FREQUENT (R2F)

### 3.1 COMPOSITIONAL TEXT-TO-IMAGE GENERATION FOR RARE CONCEPTS

**Problem Setup.** Consider a T2I generation model $\theta$ trained on data distribution $p_{\text{data}}$ involving two types of data; (i) image data, denoted by $\boldsymbol{x} \in \mathbb{R}^d$; and (ii) text prompt, denoted by $\boldsymbol{c}$. Following the compositional generation literature (Liu et al., 2022), we suppose that a text prompt consists of *a composition of concepts* as $\boldsymbol{c} = \{c_1, \ldots, c_n\}$, where each $c_i$ denotes a unit of concept, e.g., object, color, etc. For simplicity, we denote 'composition of concepts' as 'concept' hereafter. Because real-world T2I datasets typically exhibit a *long-tailed* nature (Xu et al., 2023), we naturally assume the existence of a set of rare concepts $\mathcal{C}_R$ and a set of frequent concepts $\mathcal{C}_F$, where $p_{\text{data}}(\boldsymbol{c}_F) \gg p_{\text{data}}(\boldsymbol{c}_R)$, and $p_{\text{data}}(\boldsymbol{c}_R) \approx 0, \forall \boldsymbol{c}_F \in \mathcal{C}_F$ and $\boldsymbol{c}_R \in \mathcal{C}_R$. Then, the *compositional T2I generation for rare concepts* problem is to find an approach $\phi$ that maximizes the following T2I alignment objective:

$$\operatorname{argmax}_\phi \ \mathbb{E}_{\boldsymbol{c}_R \in \mathcal{C}_R} \big[ \text{T2I-alignment}(\phi(\boldsymbol{c}_R; \theta), \boldsymbol{c}_R) \big]. \tag{1}$$

**Theoretical Motivation.** To analyze how relevant frequent concepts help the rare concept composition, we consider a simple setting where two texts are given: one with a rare concept $\boldsymbol{c}_R$ (e.g.,"*furry frog*"), and another with a frequent one $\boldsymbol{c}_F$ (e.g., "*furry dog*"). Let $\boldsymbol{x} \in \mathbb{R}^2$ be an image representation, where the first and second dimension represents the attribute (e.g., "*furry*") and object (e.g., "*animal*"), respectively. We assume the ground truth conditional distributions $p_{\text{data}}(\boldsymbol{x}|\boldsymbol{c}_R)$ and $p_{\text{data}}(\boldsymbol{x}|\boldsymbol{c}_F)$ follow Gaussian distributions $\mathcal{N}(\boldsymbol{\mu}_R, \boldsymbol{\Sigma}_R)$ and $\mathcal{N}(\boldsymbol{\mu}_F, \boldsymbol{\Sigma}_F)$ (Liang et al., 2024), where $\boldsymbol{\Sigma}_R = \boldsymbol{\Sigma}_F = \boldsymbol{I}_2$

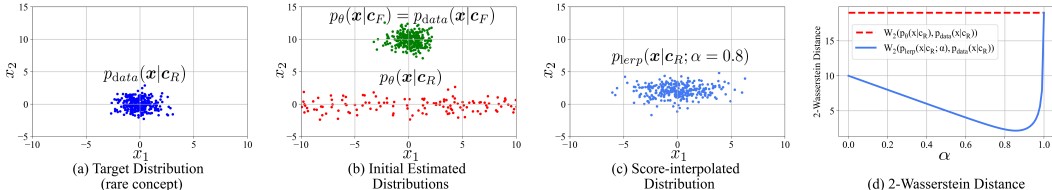

Figure 3: **Visualizing distributions in rare concept generation.** (a) The true data distribution conditioned on the rare concept $c_R$ (e.g., ("furry", "frog")), modeled as $\mathcal{N}((0,0), I_2)$; (b) Initial estimated distribution for the rare concept $c_R$, $\mathcal{N}((0,0), \text{diag}(20^2, 1))$, with high uncertainty along $x_1$ (red). The green points represent the estimated distribution for the frequent concept $c_F$ (e.g., ("furry", "dog")), $\mathcal{N}((0, 10), I_2)$; (c) The distribution generated via linear interpolation of the score functions, $p_{\text{lerp}}(\boldsymbol{x}|c_R; \alpha = 0.8)$, which combines information from both the rare and frequent concepts, yielding better approximation of the rare concept; (d) 2-Wasserstein distance between the $p_{\text{lerp}}(\boldsymbol{x}|c_R; \alpha)$ and the target distribution (blue line). The distance shows that a well-chosen $\alpha$ improves the approximation compared to using only the rare concept score function (red dashed line).

and the first component of $\boldsymbol{\mu}_R$ and $\boldsymbol{\mu}_F$ being identical. Here, $I_2$ denotes the $2 \times 2$ identity matrix. Let $\nabla_{\boldsymbol{x}} \log p_\theta(\boldsymbol{x}|c_R)$ be the estimated score function prameterized by $\theta$. We assume that the score estimator is given for each concept as: $\nabla_{\boldsymbol{x}} \log p_\theta(\boldsymbol{x}|c_F) = \nabla_{\boldsymbol{x}} \log \mathcal{N}(\boldsymbol{\mu}_F, \boldsymbol{\Sigma}_F) = \nabla_{\boldsymbol{x}} \log p_{\text{data}}(\boldsymbol{x}|c_F)$; and $\nabla_{\boldsymbol{x}} \log p_\theta(\boldsymbol{x}|c_R) = \nabla_{\boldsymbol{x}} \log \mathcal{N}(\hat{\boldsymbol{\mu}}_R, \hat{\boldsymbol{\Sigma}}_R)$, where $\hat{\boldsymbol{\mu}}_R = \boldsymbol{\mu}_R$ and $\hat{\boldsymbol{\Sigma}}_R = diag(\sigma, 1)$. Here, $diag(\sigma, 1)$ denotes the $2 \times 2$ diagonal matrix with $\sigma$ and 1 as its diagonal elements.

This setting reflects a common scenario in real-world datasets where image representations for a frequent concept are well-represented, allowing the model to learn their distributions accurately. As a result, the score estimator $\nabla_{\boldsymbol{x}} \log p_\theta(\boldsymbol{x}|c_F)$ closely matches the true conditional distribution $\nabla_{\boldsymbol{x}} \log p_{\text{data}}(\boldsymbol{x}|c_F)$. On the other hand, when generating image representations for a rare concept, the model has significantly fewer or no examples to learn from. This limited exposure increases the uncertainty in the model's predictions, leading to higher randomness. This is reflected in the score estimator for $c_R$, where $\hat{\boldsymbol{\Sigma}}_R = \text{diag}(\sigma, 1)$ with $\sigma \gg 1$. The large variance $\sigma$ captures the high uncertainty in attribute space due to the scarcity of data for rare concepts. See Figure 3 for examples.

In this setting, interpolating between the estimated score function for the frequent concept $\nabla_{\boldsymbol{x}} \log p_\theta(\boldsymbol{x}|c_F)$ and that of the rare concept $\nabla_{\boldsymbol{x}} \log p_\theta(\boldsymbol{x}|c_R)$ can yield a better approximation of the target distribution $\mathcal{N}(\boldsymbol{\mu}_R, \boldsymbol{\Sigma}_R)$, i.e., a smaller Wasserstein distance, than using only the estimated score function for the rare concept, which is shown in the Theorem 3.1.

**Theorem 3.1** (Improved rare concept generation via linear interpolation between score functions). *Given the above setting, consider the linear interpolated score estimator for the rare concept as:*

$$\alpha \nabla_{\boldsymbol{x}} \log p_\theta(\boldsymbol{x}|c_R) + (1-\alpha)\nabla_{\boldsymbol{x}} \log p_\theta(\boldsymbol{x}|c_F), \quad \alpha \in [0, 1]. \tag{2}$$

*This interpolated score function corresponds to the score function of the Gaussian distribution $\mathcal{N}(\boldsymbol{\mu}_{lerp}, \boldsymbol{\Sigma}_{lerp})$ where $\boldsymbol{\mu}_{lerp} = \alpha\boldsymbol{\mu}_R + (1-\alpha)\boldsymbol{\mu}_F$, and $\boldsymbol{\Sigma}_{lerp}^{-1} = \alpha\hat{\boldsymbol{\Sigma}}_R^{-1} + (1-\alpha)\boldsymbol{\Sigma}_F^{-1}$. Let $p_{lerp}(\mathbf{x}|c_R; \alpha) := \mathcal{N}(\boldsymbol{\mu}_{lerp}, \boldsymbol{\Sigma}_{lerp})$. If $\sigma \geq 1 + \sqrt{\|\boldsymbol{\mu}_F - \boldsymbol{\mu}_R\|^2 + 0.2}$, then the following inequality holds:*

$$\min_\alpha \mathcal{W}_2\big(p_{lerp}(\boldsymbol{x}|c_R; \alpha), p_{data}(\boldsymbol{x}|c_R)\big) < \mathcal{W}_2\big(p_\theta(\boldsymbol{x}|c_R), p_{data}(\boldsymbol{x}|c_R)\big), \tag{3}$$

*where $\mathcal{W}_2(p, q)$ denotes the 2-Wasserstein distance between the distributions $p$ and $q$.*

*Proof.* The complete proof is available in Appendix A. □

## 3.2 PROPOSED FRAMEWORK: R2F

Inspired by the theory, we propose a training-free framework, **R2F**, which leverages LLM to find frequent concepts relevant to rare concepts and use them in diffusion sampling. R2F involves a two-stage process: (i) *Rare-to-frequent concept mapping* that uses LLM to identify rare concepts and their relevant yet frequent concepts; and (ii) *Alternating concept guidance* that iteratively uses prompts involving either rare or frequent objects during the sampling process. Unlike Theorem 3.1, in multi-step denoising, we use alternating guidance as the default approach for rare-frequent interpolation (See Section 4.4 for its efficacy over interpolation). For the acceleration method with short sampling steps, we use interpolation (See Appendix K). Figure 4 illustrates the overview of our framework.

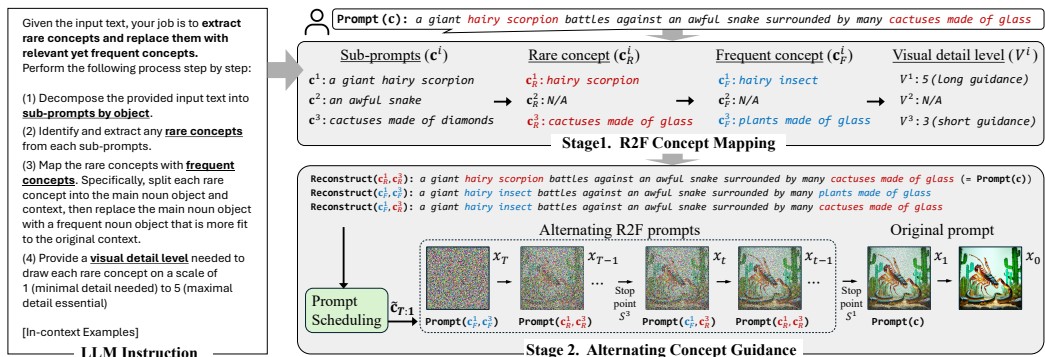

Figure 4: Overview of our R2F framework.

**Rare-to-Frequent Concept Mapping.** To extract proper concept mapping, we use LLM instruction with chain-of-thought prompting (See Appendix B for the full LLM instruction and in-context learning examples). Given a text prompt $\mathbf{c}$, R2F performs the concept mapping process step by step: (1) It decomposes the prompt into $m$ sub-prompts per object, $\mathbf{c} = \{\mathbf{c}^i\}_{i=1}^m$. For each sub-prompt $\mathbf{c}^i$, (2) it identifies rare concepts and (3) extracts a frequent concept contextually relevant to the rare concept, making a rare-to-frequent concept mapping $\hat{\mathbf{c}}^i = (\mathbf{c}^i, \mathbf{c}_R^i, \mathbf{c}_F^i)$. Specifically, to find the frequent concept $\mathbf{c}_F^i$, it splits each rare concept into the main noun object and its attributes and then replaces the main noun object into another object contextually more fit to the attributes. Additionally, (4) It extracts a visual detail level $V^i$ (an integer value from 1 to 5) needed to draw each rare concept for determining an appropriate stop point of concept guidance, based on prior observation (Hertz et al., 2022) that rough visual features (e.g., shape) are highly affected by diffusion latents at early sampling steps while detailed visual features (e.g., texture) are influenced at the later sampling steps. The final mapped rare-to-frequent concept output is formed as $\hat{\mathbf{c}} = \{(\mathbf{c}^i, \mathbf{c}_R^i, \mathbf{c}_F^i, V^i)\}_{i=1}^m$.

**Alternating Concept Guidance.** Based on the extracted concept mappings, R2F guides the diffusion inference by alternately exposing rare and frequent concepts throughout $T$ sampling steps. We first construct a scheduled batch of prompts $\tilde{\mathbf{c}}_{T:1} = \{\tilde{\mathbf{c}}_T, \ldots, \tilde{\mathbf{c}}_1\}$, which are sequentially inputted to the diffusion model. In early steps, $\tilde{\mathbf{c}}_t$ is reconstructed from either the set of frequent concept $\{\mathbf{c}_F^i\}_{i=1}^m$ or that of rare concept $\{\mathbf{c}_R^i\}_{i=1}^m$, alternatively; $\tilde{\mathbf{c}}_t = \text{Reconstruct}(\{\mathbf{c}_F^i\}_{i=1}^m)$ if $(T-t)\%2 = 0$, and $\tilde{\mathbf{c}}_t = \text{Reconstruct}(\{\mathbf{c}_R^i\}_{i=1}^m)$ otherwise. The reconstruction process is simply done by substituting the words of the detected concept from the original prompt (See $\text{Reconstruct}(\mathbf{c}_R^1, \mathbf{c}_R^3)$ in Figure 4).

Meanwhile, each frequent concept $\mathbf{c}_F^i$ stops being used to reconstruct at different stop points $S^i$ determined by the visual detail level $V^i$. The stop point $S^i$ is obtained by converting a visual detail score $V^i$ in the integer grid $[1, 2, 3, 4, 5]$ into a mapped float value in a grid $[0.9, 0.8, 0.6, 0.4, 0.2]$ and multiply it by the total diffusion step $T$; if $V^i = 1$, then $S^i = \lfloor 0.9T \rfloor$. After the stop point $S^i$, the $i$-th frequent concept is no longer used for reconstruction, so it becomes $\tilde{\mathbf{c}}_t = \text{Reconstruct}(\{\mathbf{c}_F^i\}_{i=1}^m \setminus \mathbf{c}_F^i \cup \mathbf{c}_R^i)$. This process repeats until all stop points have been passed. After the latest stop point $S^{last}$, the alternating guidance is finished and only the original prompt is inputted; $\tilde{\mathbf{c}}_t = \mathbf{c}$ when $t < S^{last}$. Each reconstructed prompt $\tilde{\mathbf{c}}_t$ in the scheduled prompt batch $\tilde{\mathbf{c}}_{T:1}$ is sequentially inputted to generate the diffusion latent $\mathbf{z}_t$, as $\mathbf{z}_{t-1} \leftarrow p_\theta(\mathbf{z}_t, \tilde{\mathbf{c}}_t)$, leading to the final output image $\mathbf{x}_0 = \text{vae}(\mathbf{z}_0)$.

### 3.3 Extention with Region-guided Diffusion Models: R2F+

We extend our framework to the general region-guided diffusion approach. The extended framework, named **R2F+**, has more controllability in generating all rare concepts in the prompt by seamlessly applying the rare-to-frequent concept guidance for each object at the appropriate region. We employ LLMs to both identify the object's appropriate region and its rare-to-frequent concept mapping. As illustrated in Figure 5, R2F+ involves a three-stage process: (i) *Region-aware rare-to-frequent concept mapping* that extracts *per-object* rare-to-frequent concept mapping with its proper position (e.g., bounding box) via LLMs, (ii) *Masked latent generation via object-wise R2F guidance* that individually generates per-object rare concept images from each sub-prompt and saves its masked latents, and (iii) *Region-controlled alternating concept guidance* that seamlessly generate compositional images from the prompt layout using masked latent fusion and cross-attention region control. More detailed LLM instructions and algorithm pseudocode for R2F+ are elaborated in Appendix C.

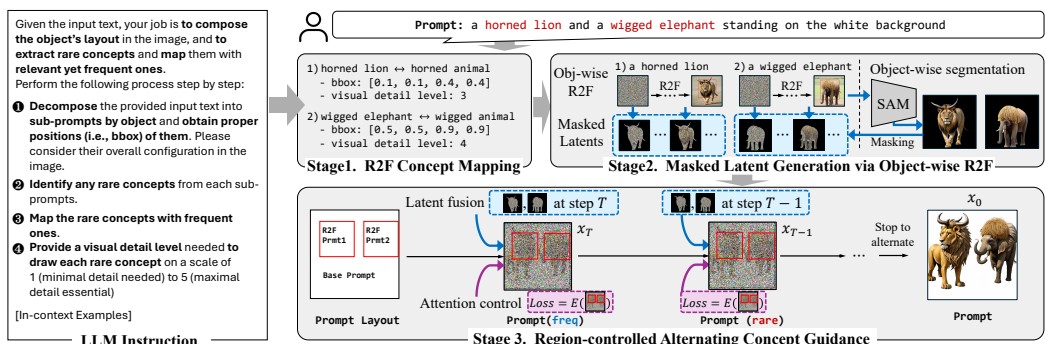

Figure 5: Overview of R2F+, the extension of our framework with region-guided diffusion.

**Region-aware Rare-to-frequent Concept Mapping.** R2F+ uses a similar chain-of-thought LLM instruction with R2F, while it also extracts each object's proper position, i.e., bounding box. Then, the region-aware rare-to-frequent concept mapping process converts the input prompt $\{\mathbf{c}^i\}_{i=1}^m$ into region-aware prompt $\{(\mathbf{c}^i, \mathbf{c}_R^i, \mathbf{c}_F^i, V^i, R^i)\}_{i=1}^m$, where $R_i$ refers to the obtained region of object $i$.

**Masked Latent Generation via Object-wise R2F.** To synthesize each object more accurately, inspired by Lian et al. (2023), we first apply the image generation process of R2F for each object individually. This is to obtain the per-object diffusion latent $\bar{\mathbf{z}}^i$ and integrate them into the final generation process with multi-objects. We generate each image of sub-prompt $(\mathbf{c}^i, \mathbf{c}_R^i, \mathbf{c}_F^i, V^i)$ with R2F guidance, and obtain all the latent $\bar{\mathbf{z}}_{T:0}^i$ throughout the sampling step $t$. Next, we adopt external object detection and segmentation models, such as SAM (Kirillov et al., 2023), to extract per-object binary mask $M^i$. The mask and latent are further refined by adjusting the size and location offset between the object-wise image and the region $R^i$, resulting in a refined mask $M^i \leftarrow \text{REFINEMASK}(M^i, R^i)$ and a refined latent $\bar{\mathbf{z}}_{T:0}^i \leftarrow \text{REFINELATENTS}(\bar{\mathbf{z}}_{T:0}^i, R^i)$.

**Region-controlled Alternating Concept Guidance.** With the obtained per-object rare-to-frequent prompts and their masked latents, we finally generate the whole objects all at once via region-controlled alternating concept guidance. To force each object to appear in the proper region, we adopt two popular region control techniques: (1) cross-attention control (Chen et al., 2024c), and (2) latent fusion (Lian et al., 2023). At each step $t$, the latent $\mathbf{z}_t$ is first modified by the cross-attention control: $\mathbf{z}_t' \leftarrow \mathbf{z}_t - \eta \nabla_{\mathbf{z}_t} \sum_{i=1}^m E(A^i, R^i)$ *(cross-attention control)*, where $A^i$ is an average cross-attention map for the object $i$ across the diffusion layers, $E(A, R) = \left(1 - \sum_{(x,y) \in R} A_{x,y} / \sum_{(x,y)} A_{x,y}\right)^2$ is an energy function, and $\eta$ is a controlling parameter. Next, the original diffusion sampling is performed, as $\mathbf{z}_{t-1} \leftarrow p_\theta(\mathbf{z}_t', \tilde{\mathbf{c}}^t)$. Afterward, we compound the controlled latent $\mathbf{z}_{t-1}$ with every object-wise latent $\bar{\mathbf{z}}_{t-1}^i$ sequentially via the latent fusion: $\mathbf{z}_{t-1} \leftarrow \mathbf{z}_{t-1} \odot (1 - M^i) + \bar{\mathbf{z}}_{t-1}^i \odot M^i$ *(latent fusion)*. Both the cross-attention control and the latent fusion are applied only to initial sampling steps until the stop points, integrated with the alternating rare-to-frequent concept guidance.

## 4 EXPERIMENTS

### 4.1 EXPERIMENT SETTING

**Datasets.** We evaluate R2F using one new dataset, **RareBench**, which we design to validate the generated image quality for *rare* concepts, and two existing datasets, DVMP (Rassin et al., 2024) and T2I-CompBench (Huang et al., 2023), for general image compositionally. As shown in Table 1, RareBench includes more rare concepts compared to existing benchmarks.

1. **RareBench**: We design our benchmark to contain prompts with *various* rare concept across single- and multi-objects. For single-object prompts, we categorize the rare concept attributes with *five* cases: property ("a *hairy* frog"), shape ("a *banana-shaped* apple"), texture ("a *tiger-stripped* lizard"), action ("a *yawning* crab"), and complex ("a *rainbow* elephant *spitting fire*"). For multi-object prompts, we further introduce *three* more cases: concatenation ("a *hairy* shark and two *wigged* octopuses"), relation ("a *thorny* snake *is coiling around* a *star-shaped* drum"), and a more complex case with multi-objects. We generate 40 prompts for each of the aforementioned eight cases, constructing a total of 320 prompts. To ensure the prompt's rareness, we use GPT to make prompts contain contextually rare attribute-object pairs, which is detailed in Appendix D.

Table 2: Text-to-image alignment performances of R2F and other baselines on the **RareBench** dataset. The best values are in blue and the second best values are in green .

| Models | Property | | Shape | | Single Object Texture | | Action | | Complex | | Concat | | Multi Objects Relation | | Complex | |
|---|---|---|---|---|---|---|---|---|---|---|---|---|---|---|---|---|
| | GPT4 | Human | GPT4 | Human | GPT4 | Human | GPT4 | Human | GPT4 | Human | GPT4 | Human | GPT4 | Human | GPT4 | Human |
| SD1.5 | 55.0 | 49.6 | 38.8 | 51.7 | 33.8 | 55.6 | 23.1 | 47.5 | 36.9 | 44.2 | 23.1 | 29.8 | 24.4 | 20.0 | 36.3 | 19.8 |
| SDXL | 60.0 | 55.2 | 56.9 | 57.7 | 71.3 | 63.3 | 47.5 | 59.0 | 58.1 | 60.4 | 39.4 | 35.8 | 35.0 | 28.8 | 47.5 | 41.7 |
| PixArt | 49.4 | 59.6 | 58.8 | 60.8 | 76.9 | 69.0 | 56.3 | 69.8 | 63.1 | 70.6 | 35.6 | 38.1 | 30.0 | 31.0 | 48.1 | 42.7 |
| SD3.0 | 49.4 | 66.9 | 76.3 | 79.0 | 53.1 | 62.7 | 71.9 | 73.3 | 65.0 | 70.8 | 55.0 | 64.6 | 51.2 | 55.2 | 70.0 | 63.5 |
| FLUX | 58.1 | 63.8 | 71.9 | 70.0 | 47.5 | 61.7 | 52.5 | 67.1 | 60.0 | 67.3 | 55.0 | 57.3 | 48.1 | 50.6 | 70.3 | 66.7 |
| SynGen | 61.3 | 46.9 | 59.4 | 44.8 | 54.4 | 57.3 | 33.8 | 48.3 | 50.6 | 49.0 | 30.6 | 35.8 | 33.1 | 23.5 | 29.4 | 20.4 |
| LMD | 23.8 | 41.5 | 35.6 | 46.0 | 27.5 | 51.5 | 23.8 | 45.2 | 35.6 | 39.8 | 33.1 | 23.5 | 34.4 | 30.4 | 33.1 | 21.0 |
| RPG | 33.8 | 47.1 | 54.4 | 57.1 | 66.3 | 60.8 | 31.9 | 44.0 | 37.5 | 38.1 | 21.9 | 25.6 | 15.6 | 14.4 | 29.4 | 39.6 |
| ELLA | 31.3 | 49.6 | 61.6 | 54.8 | 64.4 | 61.9 | 43.1 | 53.8 | 66.3 | 60.6 | 42.5 | 45.6 | 50.6 | 39.6 | 51.9 | 47.9 |
| **R2F** | 89.4 | 86.3 | 79.4 | 80.6 | 81.9 | 71.5 | 80.0 | 79.4 | 72.5 | 75.6 | 70.0 | 71.3 | 58.8 | 57.9 | 73.8 | 67.3 |

2. **DVMP**: This benchmark includes prompts that randomly bind 38 objects with 26 attributes (13 of them are colors). We divide it into single- and multi-object cases, each of which has 100 prompts.

3. **T2I-CompBench**: This benchmark consists of compositional prompts with multi-objects. The benchmark covers various concepts to validate compositional generation, while relatively common.

**Implementation Details.** Our R2F is *flexible* to an *arbitrary* combination of diffusion models and LLMs. By default, we use SD3.0 and GPT-4o (Achiam et al., 2023). For every inference, we set the sampling steps $T$ to 50 and the random seed to 42. The hyperparameters for all baselines are favorably configured following the original papers. All methods are implemented with PyTorch 2.0.0 and executed on NVIDIA A100 GPUs. See Appendix O for GPU efficiency analysis.

Table 1: Dataset statistics. %Rareness is calculated by asking GPT4 if each prompt contains rare concepts (See Appendix E for calculation details).

| Datasets | RareBench | DVMP | T2I-CompBench |
|---|---|---|---|
| # Prompts | 320 | 200 | 2400 |
| % Rareness | 98.1 | 62.0 | 17.4 |

**Baselines.** We compare R2F with *nine* existing approaches of three types: (1) pre-trained T2I diffusion models; SD1.5 (Rombach et al., 2022), SDXL (Podell et al., 2023), PixArt-$\alpha$ (Chen et al., 2023), SD3.0 (Esser et al., 2024), and FLUX-schnell (BlackForestLabs, 2024), (2) a region-controlled approach; SynGen (Rassin et al., 2024), and (3) LLM-grounded diffusion models; LMD (Lian et al., 2023), RPG (Yang et al., 2024), and ELLA (Hu et al., 2024) which are built on SDXL.

**Evaluation.** We evaluate T2I-alignment by GPT-4o and humans. For precise evaluation, we ask GPT-4o and humans with a detailed score rubric (Chen et al., 2024b), which are detailed in Appendix F.

## 4.2 MAIN RESULTS OF R2F

**RareBench.** Table 2 shows the T2I alignment performance of R2F compared to all the baselines on RareBench. Overall, R2F consistently performs the best in both GPT-4o and Human evaluations across all cases of rare concepts. Numerically, R2F outperforms the best baselines for each case from $3.1\%p$ to $28.1\%p$ in GPT-4o evaluation and from $0.6\%p$ to $19.4\%p$ in Human evaluation. Among the baselines, the latest pre-trained diffusion model, SD3.0, tends to achieve higher alignment scores possibly because of the advanced training technique. Interestingly, the *regional* LLM-grounded diffusion models, LMD and RPG, show worse results than their backbone diffusion SDXL on RareBench, as they are designed to generate objects in specific regions and not designed to tightly bind rare attributes to the object in the same region. Figure 6 illustrates the generated image examples. R2F succeeds in generating images of diverse cases of rare concepts, while maintaining the realism of the images. More visualization results with varying random seeds are in Appendix G, and the rare-to-frequent concept mapping examples extracted by R2F are in Appendix H.

**DVMP and T2I-CompBench.** Table 3 summarizes the T2I alignment performance of the diffusion models on DVMP and T2I-CompBench. For DVMP, similar to the result on RareBench, R2F consistently performs the best in both GPT-4o and human evaluation. For T2I-CompBench, while R2F performs the best in GPT-4o evaluation, it performs the secondary which may be because the T2I-CompBench's auto evaluation metrics, i.e., BLIP, UniDet, etc, are sometimes showing inaccurate results (See Appendix I). Numerically, with the GPT-4o evaluation, R2F outperforms the best baselines from $2.7\%p$ to $5.5\%p$ on DVMP and from $0.1\%p$ to $3.6\%p$ on T2I-CompBench. The slightly lower improvements compared to the results on RareBench indicate that the effectiveness of R2F is proportional to the prompt rareness of the dataset, as shown in Table 1.

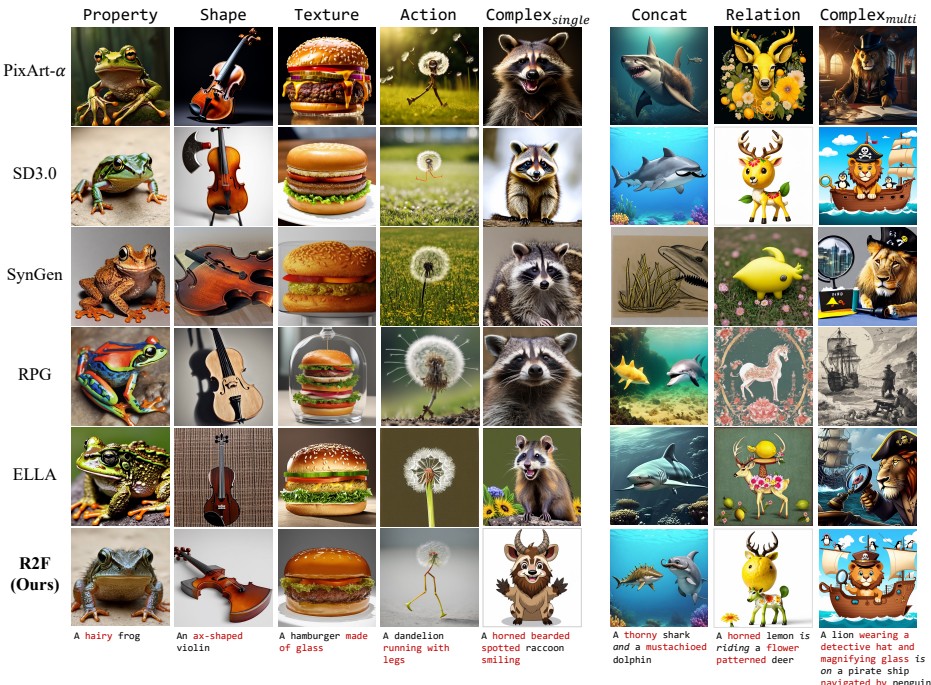

Figure 6: Qualitative comparison of R2F with state-of-the-art diffusion baselines on RareBench.

Table 3: T2I alignment performance of R2F and diffusion baselines on DVMP and T2I-CompBench. The values with the marking † are from (Yang et al., 2024).

| Models | DVMP | | | | T2I-CompBench | | | | | | | | | | |
| | Single | | Multi | | Color | | Shape | | Texture | | Spatial | | Non-Spatial | | Complex | |
| | GPT4 | Human | GPT4 | Human | GPT4 | BLIP | GPT4 | BLIP | GPT4 | BLIP | GPT4 | UniDet | GPT4 | CLIP | GPT4 | 3in1 |
|---|---|---|---|---|---|---|---|---|---|---|---|---|---|---|---|---|
| SD1.5 | 65.0 | 67.6 | 37.8 | 34.4 | 50.2 | 37.5 | 37.8 | 38.8 | 57.5 | 44.1 | 34.8 | 9.5 | 81.9 | 31.2 | 60.9 | 30.8 |
| SDXL | 76.8 | 67.1 | 59.0 | 52.6 | 68.3 | 63.7† | 53.0 | 54.1† | 72.7 | 56.4† | 55.3 | 20.3† | 83.8 | 31.1† | 70.6 | 40.9† |
| PixArt | 73.5 | 78.3 | 44.0 | 44.9 | 51.4 | 68.9† | 39.2 | 55.8† | 65.0 | 70.4† | 43.7 | 20.8† | 87.3 | 31.8† | 70.4 | 41.2† |
| SD3.0 | 74.0 | 79.0 | 72.5 | 72.3 | 90.3 | 84.0 | 76.2 | 63.3 | 91.3 | 80.1 | 72.0 | 34.0 | 88.5 | 31.4 | 85.2 | 47.7 |
| FLUX | 66.8 | 73.8 | 72.5 | 71.8 | 88.7 | 82.5 | 71.2 | 59.4 | 90.0 | 78.1 | 73.0 | 35.0 | 88.7 | 31.7 | 83.4 | 45.5 |
| SynGen | 74.5 | 60.0 | 57.0 | 41.1 | 72.8 | 70.0 | 51.5 | 45.5 | 79.6 | 60.1 | 48.5 | 22.6 | 72.8 | 31.0 | 63.2 | 33.3 |
| LMD | 47.8 | 52.5 | 45.5 | 45.1 | 62.6 | 65.3 | 61.3 | 56.9 | 74.3 | 54.7 | 73.7 | 34.3 | 51.2 | 30.0 | 62.2 | 34.2 |
| RPG | 74.0 | 65.4 | 30.0 | 27.6 | 80.5 | 83.4† | 74.2 | 68.0† | 82.2 | 81.3† | 65.3 | 45.5† | 88.6 | 34.6† | 81.0 | 54.1† |
| ELLA | 68.3 | 71.1 | 61.3 | 56.1 | 81.5 | 78.2 | 61.6 | 58.8 | 80.8 | 71.3 | 53.5 | 29.6 | 81.0 | 31.5 | 76.8 | 44.4 |
| **R2F** | 79.5 | 82.0 | 78.0 | 74.6 | 90.5 | 84.3 | 77.6 | 63.9 | 91.9 | 81.7 | 75.6 | 45.6 | 89.2 | 32.5 | 85.3 | 47.9 |

Table 4: Performance of R2F combined with different diffusion models (SDXL, IterComp, and SD3.0) on RareBench.

| Models | Single Object | | | | | Multi Objects | | |
| | Property | Shape | Texture | Action | Complex | Concat | Relation | Complex |
|---|---|---|---|---|---|---|---|---|
| SDXL | 60.0 | 56.9 | 71.3 | 47.5 | 58.1 | 39.4 | 35.0 | 47.5 |
| LMD | 23.8 | 35.6 | 27.5 | 23.8 | 35.6 | 33.1 | 34.4 | 33.1 |
| RPG | 33.8 | 54.4 | 66.3 | 31.9 | 37.5 | 21.9 | 15.6 | 9.4 |
| ELLA | 31.3 | 63.6 | 64.4 | 43.1 | 66.3 | 42.5 | **50.6** | 51.9 |
| **R2F_sdxl** | **71.3** | **71.9** | **73.8** | **54.4** | **70.6** | **50.6** | 36.0 | **52.8** |
| IterComp | 63.8 | 66.9 | 61.3 | 65.6 | 61.9 | 41.3 | 29.4 | 53.1 |
| **R2F_itercomp** | **78.1** | **77.5** | **79.4** | **66.9** | **63.9** | **41.5** | **36.6** | **53.4** |
| SD3.0 | 49.4 | 76.3 | 53.1 | 71.9 | 65.0 | 55.0 | 51.2 | 70.0 |
| **R2F_sd3.0** | **89.4** | **79.4** | **81.9** | **80.0** | **72.5** | **70.0** | **58.8** | **73.8** |

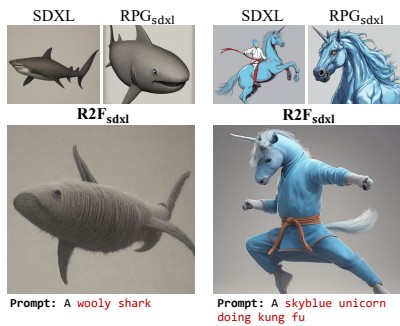

Figure 7: Generated images by R2F_sdxl.

## 4.3 FLEXIBILITY ACROSS VARIOUS DIFFUSION MODELS AND LLMS

**Effectiveness across Different Diffusion Models.** Table 4 shows GPT-4o evaluated T2I alignment performance of R2F combined with various diffusion backbones, including SDXL, IterComp, and SD3.0. R2F consistently improves the performance of three backbones in all cases, regardless of the backbone used. Also, R2F_sdxl outperforms existing LLM-grounded diffusion baselines based on SDXL in most cases, indicating its effectiveness in leveraging relevant frequent concepts in generating images of rare concepts. Figure 7 visualizes the generated images by R2F_sdxl.

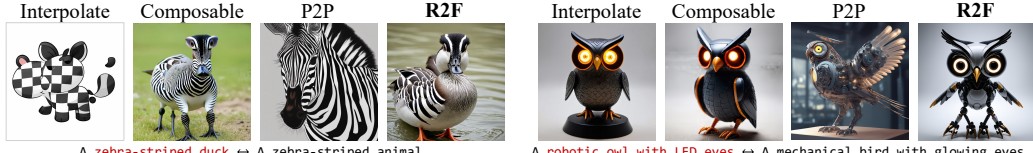

Figure 8: Qualitative comparison of R2F's *alternating* guidance with other possible guidance choices (Composable Diffusion (Liu et al., 2022) and Promt-to-Prompt (P2P) (Hertz et al., 2022)).

Table 6: Quantitative comparison of R2F's alternating guidance compared to other possible guidance choices.

| Models | Single Object | | | | | Multi Objects | | |
|---|---|---|---|---|---|---|---|---|
| | Property | Shape | Texture | Action | Complex | Concat | Relation | Complex |
| SD3.0 | 49.4 | 76.3 | 53.1 | 71.9 | 65.0 | 55.0 | 51.2 | 70.0 |
| Interpolate | 85.5 | 77.0 | 69.4 | 74.6 | 71.7 | 54.0 | 53.8 | 71.3 |
| Composable | 82.5 | 76.3 | 58.1 | 68.1 | 67.5 | 63.1 | 51.9 | 61.9 |
| P2P | 71.3 | 46.3 | 46.9 | 38.8 | 52.5 | 31.3 | 32.5 | 33.8 |
| R2F | **89.4** | **79.4** | **81.9** | **80.0** | **72.5** | **70.0** | **58.8** | **73.8** |

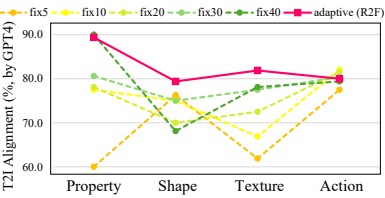

Figure 9: Efficacy of R2F's *adaptive* visual-detail-aware stop points.

**Robustness across Different LLMs.** Table 5 shows the robustness of R2F over the different LLMs including a proprietary LLM, GPT-4o, and one of the latest open-source LLM, LLaMA3-8B-Instruct (Dubey et al., 2024). R2F combined with LLaMA3 also results in substantial performance improvement compared to SD3.0, yet that combined with GPT-4o shows better results.

Table 5: Performance of R2F combined with different LLMs (GPT-4o or LLaMA3).

| Models | Property | Shape | Texture | Action | Complex |
|---|---|---|---|---|---|
| SD3.0 | 49.4 | 76.3 | 53.1 | 71.9 | 65.0 |
| R2F$_{\text{LLaMA3}}$ | 81.9 | 77.1 | 76.3 | 78.8 | 67.7 |
| R2F$_{\text{GPT-4o}}$ | 89.4 | 79.4 | 81.9 | 80.0 | 72.5 |

## 4.4 ABLATION STUDIES

**Efficacy of Alternating Guidance.** Figure 8 and Table 6 show the qualitative and quantitative analysis of the R2F's alternating guidance compared to other possible guidance choices. We apply *three* guidance choices, (1) Linear interpolation (Interpolate) of latents as in Theorem 3.1, and bring the idea of (2) Composable Diffusion (Liu et al., 2022) and (3) Prompt-to-prompt (P2P) (Hertz et al., 2022). Given a pair of rare-frequent concept prompts, Interpolate linearly interpolates the latants of rare and frequent prompts with $\alpha = 0.5$ and Composable blends the two prompt embeddings and uses it as the input, until the stop points obtained from LLM. P2P first generates a complete image from the frequent concept prompt and then edits it by the rare concept prompt with attention-control.

Overall, R2F's alternating guidance performs the best in terms of T2I alignment and image quality. This may be because linear interpolation and Composable generates images from blended latents or embeddings, which are not the real inputs that diffusion models have seen in the training phase, generating unusual images, e.g., "*A zebra-striped duck*", or blurry images, e.g., "*A robotic owl*". Also, since P2P starts editing from the complete image of the frequent concept, it tends to preserve too many features of the frequent concept when generating the original rare concept, e.g., most features of zebra are still alive even after editing it with the prompt "*A zebra-striped duck*".

**Efficacy of Visual-detail-aware Guidance Stop Points.** Figure 9 depicts the efficacy of R2F's *adaptive* visual-detail-aware stop points compared to when using a *fixed* stop point on RareBench with single-object case, which has only one stop point. We ablate the fixed stop point in the grid of $\{5, 10, 20, 30, 40\}$. With lower stop points such as 5 and 10 (in yellow lines), R2F shows relatively lower performance than those with higher stop points (in green lines) in generating rare concepts for attribute types of property and texture, because these usually require a higher level of visual details to synthesize. This tendency becomes reversed for the attribute type of shape, which tends to require a lower level of visual details. The original R2F, which adaptively determines the guidance stop points based on the appropriate visual detail level for each prompt, naturally leads to the best performance.

## 4.5 CONTROLLABLE IMAGE GENERATION RESULTS OF R2F+

Table 7 shows the superiority of R2F+ for controllable image generation on three benchmarks with multi-object cases. Overall, R2F+ mostly performs the best in terms of T2I alignment accuracy, outperforming even R2F by leveraging more detailed layout-guided image generation process as shown in Figure 10. In addition, Figure 11 visualizes the generated images of R2F+ compared to SD3.0. With the proper layouts and their rare-to-frequent concept guidance generated by LLM,

Table 7: Controllable image generation performance of R2F+ on three benchmarks with multi-object cases. T2I alignment accuracy measured from GPT4 was reported.

| Models | RareBench$_{multi}$ | | | DVMP | T2I-CompBench | | | | | |
| | Concat | Relat | Compl | Multi | Color | Shape | Textr | Spat | Non-Spat | Compl |
|---|---|---|---|---|---|---|---|---|---|---|
| SD3.0 | 55.0 | 51.2 | 70.0 | 72.5 | 90.3 | 76.2 | 91.3 | 72.0 | 88.5 | 85.2 |
| **R2F** | 70.0 | 58.8 | 73.8 | 78.0 | 90.5 | 77.6 | 91.9 | 75.6 | 89.2 | 85.3 |
| **R2F+** | 74.4 | 63.7 | 64.8 | 81.5 | 91.5 | 84.1 | 93.0 | 82.2 | 91.1 | 80.7 |

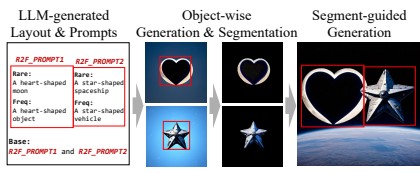

PROMPT: A heart-shaped moon and a star-shaped spaceship

Figure 10: Detailed LLM-based layout-guided generation process of R2F+.

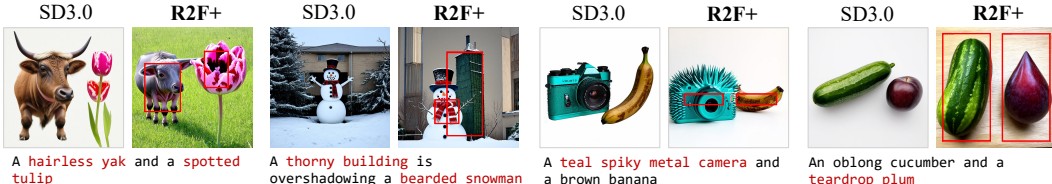

A hairless yak and a spotted tulip

A thorny building is overshadowing a bearded snowman

A teal spiky metal camera and a brown banana

An oblong cucumber and a teardrop plum

Figure 11: Qualitative comparison of SD3.0 and R2F+ on three benchmarks. Best viewed in zoom.

R2F+ consistently shows controllable image generation results, achieving proper attribute binding on the corresponding region of objects. Meanwhile, R2F+ can not accurately synthesize images from prompts with 'complex' cases where multi-objects are intertwined in the region. This may be because the training-free layout-guided generation often fails with overlapped bounding boxes (Yang et al., 2024). More analysis with failure cases and image quality scores are in Appendix J and Appendix N.

### 4.6 Superiority and Compatibility over Prompt Paraphrasing

Table 8: Superiority and compatibility of R2F over prompt paraphrasing. Best and secondary values are in bold and italics, respectively.

| Models | Property | Shape | Texture | Action | Complex |
|---|---|---|---|---|---|
| SD3.0 | 49.4 | 76.3 | 53.1 | 71.9 | 65.0 |
| SD3.0$_{paraphrase}$ | 64.4 | 65.6 | 55.6 | *85.6* | 65.6 |
| R2F | **89.4** | **79.4** | *81.9* | 80.0 | 72.5 |
| **R2F$_{paraphrase}$** | *83.1* | *73.1* | **82.0** | **86.9** | **74.0** |

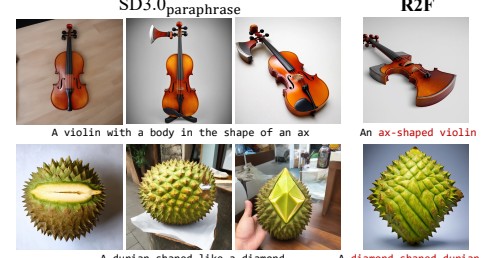

SD3.0$_{paraphrase}$ · R2F

A violin with a body in the shape of an ax · An ax-shaped violin

A durian shaped like a diamond · A diamond-shaped durian

Figure 12: Generated images with paraphrasing.

**Paraphrasing Rule.** We use GPT-4o to investigate the effect of paraphrasing in generating rare concepts by asking '*I will draw a picture based on the given CAPTION. Please paraphrase it so that I can draw it more easily, while not changing the meaning.*'.

**Results.** Table 8 shows the superiority and compatibility of R2F over prompt paraphrasing by GPT-4o. With prompt paraphrasing, the T2I alignment of SD3.0 is enhanced, but it is not as significant as R2F. Interestingly, as shown in Figure 12, some rare concepts, e.g., "*violin with ax-shaped body*" and "*diamond-shaped durian*", are extremely hard to synthesize even after a careful paraphrasing. On the other hand, R2F succeeds in synthesizing. This shows the genuine superiority of our framework in unlocking the compositional generation power of diffusion models on such rare concepts. Furthermore, R2F is compatible with paraphrasing, so it can be applied even after paraphrasing and further improve the performance.

## 5 Conclusion

In this paper, we propose R2F, a novel framework that grounds LLMs and T2I diffusion models for enhanced compositional generation on rare concepts. Based on empirical and theoretical observations that relevant frequent concepts can guide diffusion models for more accurate concept compositions, we use the LLM to extract rare-to-frequent concept mapping, and plan to route the sampling process with alternating concept guidance. Our framework is flexible across any pre-trained diffusion models and LLMs, and we further propose a seamless integration with the region-guided diffusion approaches. Extensive experiments on three datasets, including our newly proposed benchmark, RareBench, containing various prompts with rare compositions of concepts, R2F significantly outperforms existing diffusion baselines including SD3.0 and FLUX.

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

# Rare-to-Frequent: Unlocking Compositional Generation Power of Diffusion Models on Rare Concepts with LLM Guidance

## (Supplementary Material)

## A  COMPLETE PROOF OF THEOREM 3.1

**Theorem 3.1** (Improved rare concept generation via linear interpolation between score functions).
*Given the above setting, consider the linear interpolated score estimator for the rare concept as:*

$$\alpha \nabla_{\boldsymbol{x}} \log p_\theta(\boldsymbol{x}|\boldsymbol{c}_R) + (1-\alpha) \nabla_{\boldsymbol{x}} \log p_\theta(\boldsymbol{x}|\boldsymbol{c}_F), \quad \alpha \in [0,1]. \tag{2}$$

*This interpolated score function corresponds to the score function of the Gaussian distribution* $\mathcal{N}(\boldsymbol{\mu}_{lerp}, \boldsymbol{\Sigma}_{lerp})$ *where* $\boldsymbol{\mu}_{lerp} = \alpha\boldsymbol{\mu}_R + (1-\alpha)\boldsymbol{\mu}_F$, *and* $\boldsymbol{\Sigma}_{lerp}^{-1} = \alpha\hat{\boldsymbol{\Sigma}}_R^{-1} + (1-\alpha)\boldsymbol{\Sigma}_F^{-1}$. *Let* $p_{lerp}(\mathbf{x}|\boldsymbol{c}_R; \alpha) := \mathcal{N}(\boldsymbol{\mu}_{lerp}, \boldsymbol{\Sigma}_{lerp})$. *If* $\sigma \geq 1 + \sqrt{\|\boldsymbol{\mu}_F - \boldsymbol{\mu}_R\|^2 + 0.2}$, *then the following inequality holds:*

$$\min_\alpha \mathcal{W}_2\big(p_{lerp}(\boldsymbol{x}|\boldsymbol{c}_R; \alpha), p_{data}(\boldsymbol{x}|\boldsymbol{c}_R)\big) < \mathcal{W}_2\big(p_\theta(\boldsymbol{x}|\boldsymbol{c}_R), p_{data}(\boldsymbol{x}|\boldsymbol{c}_R)\big), \tag{3}$$

*where* $\mathcal{W}_2(p, q)$ *denotes the 2-Wasserstein distance between the distributions $p$ and $q$.*

*Proof.* First, we compute the 2-Wasserstein distance between two Gaussians, $p_\theta(\boldsymbol{x}|\boldsymbol{c}_R)$ and $p_{\text{data}}(\boldsymbol{x}|\boldsymbol{c}_R)$, using the closed form in Villani et al. (2009):

$$\mathcal{W}_2^2(p_\theta(\boldsymbol{x}|\boldsymbol{c}_R), p_{\text{data}}(\boldsymbol{x}|\boldsymbol{c}_R)) = \|\hat{\boldsymbol{\mu}}_R - \boldsymbol{\mu}_R\|^2 + \text{Tr}\left(\hat{\boldsymbol{\Sigma}}_R + \boldsymbol{\Sigma}_R - 2\left(\hat{\boldsymbol{\Sigma}}_R^{1/2}\boldsymbol{\Sigma}_R\hat{\boldsymbol{\Sigma}}_R^{1/2}\right)^{1/2}\right)$$

$$= \text{Tr}\left(\hat{\boldsymbol{\Sigma}}_R + \boldsymbol{\Sigma}_R - 2\left(\hat{\boldsymbol{\Sigma}}_R^{1/2}\boldsymbol{\Sigma}_R\hat{\boldsymbol{\Sigma}}_R^{1/2}\right)^{1/2}\right)$$

$$= \text{Tr}\left(\begin{pmatrix} \sigma^2 & 0 \\ 0 & 1 \end{pmatrix} + \begin{pmatrix} 1 & 0 \\ 0 & 1 \end{pmatrix} - 2\begin{pmatrix} \sigma & 0 \\ 0 & 1 \end{pmatrix}\right)$$

$$= (\sigma^2 + 1 - 2\sigma) + (1 + 1 - 2 \cdot 1)$$

$$= (\sigma - 1)^2,$$

where the second equality holds from the assumption that $\hat{\boldsymbol{\mu}}_R = \boldsymbol{\mu}_R$.

Next, we will show that the *linear interpolated* score function,

$$s_{\text{lerp}}(\boldsymbol{x}; \boldsymbol{c}_R; \alpha) := \alpha \nabla_{\boldsymbol{x}} \log p_\theta(x|\boldsymbol{c}_R) + (1-\alpha) \nabla_{\boldsymbol{x}} \log p_\theta(x|\boldsymbol{c}_F), \quad \alpha \in [0,1], \tag{4}$$

is the score function of a Gaussian distribution. Starting from the definition of Gaussian distribution, we have

$$s_{\text{lerp}}(\boldsymbol{x}; \boldsymbol{c}_R; \alpha) = \alpha \nabla_{\boldsymbol{x}} \log p_\theta(\boldsymbol{x}|\boldsymbol{c}_R) + (1-\alpha) \nabla_{\boldsymbol{x}} \log p_\theta(\boldsymbol{x}|\boldsymbol{c}_F)$$

$$= -\left(\alpha\hat{\boldsymbol{\Sigma}}_R^{-1}(\boldsymbol{x} - \boldsymbol{\mu}_R) + (1-\alpha)\boldsymbol{\Sigma}_F^{-1}(\boldsymbol{x} - \boldsymbol{\mu}_F)\right)$$

$$= -(\alpha\hat{\boldsymbol{\Sigma}}_R^{-1} + (1-\alpha)\hat{\boldsymbol{\Sigma}}_F^{-1})\boldsymbol{x} + \left(\alpha\hat{\boldsymbol{\Sigma}}_R^{-1}\hat{\boldsymbol{\mu}}_R + (1-\alpha)\boldsymbol{\Sigma}_F^{-1}\boldsymbol{\mu}_F\right)$$

$$= -\boldsymbol{\Sigma}_{\text{lerp}}^{-1}(\boldsymbol{x} - \boldsymbol{\mu}_{\text{lerp}}),$$

where $\boldsymbol{\Sigma}_{lerp}^{-1} := \alpha\hat{\boldsymbol{\Sigma}}_R^{-1} + (1-\alpha)\boldsymbol{\Sigma}_F^{-1}$ and $\boldsymbol{\mu}_{lerp} := \boldsymbol{\Sigma}_{lerp}\left(\alpha\hat{\boldsymbol{\Sigma}}_R^{-1}\boldsymbol{\mu}_R + (1-\alpha)\boldsymbol{\Sigma}_F^{-1}\boldsymbol{\mu}_F\right)$.

$s_{\text{lerp}}(\boldsymbol{x}; \boldsymbol{c}_R; \alpha)$ is the score function of the Gaussian distribution $p_{\text{lerp}}(\boldsymbol{x}|\boldsymbol{c}_R; \alpha) := \mathcal{N}(\boldsymbol{\mu}_{\text{lerp}}, \boldsymbol{\Sigma}_{\text{lerp}}; \alpha)$:

$$\boldsymbol{\Sigma}_{\text{lerp}}^{-1} := \alpha\hat{\boldsymbol{\Sigma}}_R^{-1} + (1-\alpha)\boldsymbol{\Sigma}_F^{-1} = \text{diag}\left(\frac{\alpha + (1-\alpha)\sigma^2}{\sigma^2}, 1\right);$$

$$\boldsymbol{\mu}_{\text{lerp}} := \alpha\boldsymbol{\mu}_R + (1-\alpha)\boldsymbol{\mu}_F,$$

assuming the first component of $\boldsymbol{\mu}_R$ and $\boldsymbol{\mu}_F$ is identical.

Now, we can compute the 2-Wasserstein distance between $p_{\text{lerp}}(\boldsymbol{x}|\boldsymbol{c}_R; \alpha)$ and $p_{\text{data}}(\boldsymbol{x}|\boldsymbol{c}_R)$:

$$
\mathcal{W}_2^2(p_{\text{lerp}}(\boldsymbol{x}|\boldsymbol{c}_R; \alpha), p_{\text{data}}(\boldsymbol{x}|\boldsymbol{c}_R))
$$

$$
= \|\boldsymbol{\mu}_{\text{lerp}} - \boldsymbol{\mu}_R\|^2 + \text{Tr}\left(\boldsymbol{\Sigma}_{\text{lerp}} + \boldsymbol{\Sigma}_R - 2\left(\boldsymbol{\Sigma}_{\text{lerp}}^{1/2}\boldsymbol{\Sigma}_R\boldsymbol{\Sigma}_{\text{lerp}}^{1/2}\right)^{1/2}\right)
$$

$$
= \|(1-\alpha)(\boldsymbol{\mu}_F - \boldsymbol{\mu}_R)\|^2 + \text{Tr}\left(\left(\begin{matrix} \frac{\sigma^2}{(1-\alpha)\sigma^2+\alpha} & 0 \\ 0 & 1 \end{matrix}\right) + \left(\begin{matrix} 1 & 0 \\ 0 & 1 \end{matrix}\right) - 2\left(\begin{matrix} \frac{\sigma}{\sqrt{(1-\alpha)\sigma^2+\alpha}} & 0 \\ 0 & 1 \end{matrix}\right)\right)
$$

$$
= \|(1-\alpha)(\boldsymbol{\mu}_F - \boldsymbol{\mu}_R)\|^2 + \left(\frac{\sigma}{\sqrt{(1-\alpha)\sigma^2+\alpha}} - 1\right)^2
$$

$$
\leq \|(1-\alpha)(\boldsymbol{\mu}_F - \boldsymbol{\mu}_R)\|^2 + \left(\frac{\sigma}{\sqrt{(1-\alpha)\sigma^2}} - 1\right)^2
$$

$$
= \|(1-\alpha)(\boldsymbol{\mu}_F - \boldsymbol{\mu}_R)\|^2 + \left(\frac{1}{\sqrt{1-\alpha}} - 1\right)^2.
$$

Therefore, if $\sigma \geq 1 + \sqrt{\|\boldsymbol{\mu}_F - \boldsymbol{\mu}_R\|^2 + 0.2}$

$$
\min_{\alpha} \mathcal{W}_2^2(p_{\text{lerp}}(\boldsymbol{x}|\boldsymbol{c}_R; \alpha), p_{\text{data}}(\boldsymbol{x}|\boldsymbol{c}_R)) \leq \min_{\alpha} \|(1-\alpha)(\boldsymbol{\mu}_F - \boldsymbol{\mu}_R)\|^2 + \left(\frac{1}{\sqrt{1-\alpha}} - 1\right)^2
$$

$$
\leq \min_{\alpha} \|(\boldsymbol{\mu}_F - \boldsymbol{\mu}_R)\|^2 + \left(\frac{1}{\sqrt{1-\alpha}} - 1\right)^2
$$

$$
\leq \|(\boldsymbol{\mu}_F - \boldsymbol{\mu}_R)\|^2 + \left(\frac{1}{\sqrt{1-0.5}} - 1\right)^2
$$

$$
\leq \|(\boldsymbol{\mu}_F - \boldsymbol{\mu}_R)\|^2 + 0.2
$$

$$
\leq (\sigma - 1)^2
$$

$$
= \mathcal{W}_2^2(p_\theta(\boldsymbol{x}|\boldsymbol{c}_R), p_{\text{data}}(\boldsymbol{x}|\boldsymbol{c}_R)).
$$

This completes the proof. $\qquad\square$

## B  LLM INSTRUCTION FOR R2F

Table 9 and Table 10 detail the full LLM prompt and the in-context examples for R2F, respectively. With the detailed chain-of-thought instruction, we can automatically find rare concepts, generate related frequent concepts, and extract the visual detail level of concepts in *one-shot* LLM inference.

Table 9: Full LLM instruction for R2F to generate rare-to-frequent concept mappings.

**\<System Prompt\>**
You are a helper language model for a text-to-image generation program that aims to create images based on input text. The program often struggles to accurately generate images when the input text contains rare concepts that are not commonly found in reality. To address this, when a rare concept is identified in the input text, you should replace it with relevant yet more frequent concepts.

**\<User Prompt\>**
Extract rare concepts from the input text and replace them with relevant yet more frequent ones. Perform the following process step by step:
  a. Identify and extract any rare concepts from the provided input text. If the text contains one or more rare concepts, extract them all. If there are no rare concepts present, do not extract any concepts. The extracted rare concepts should not overlap.
  b. Given the rare concepts, replace each extracted rare concept with a more frequent concept. Specifically, split each rare concept into the main noun subject and the context, and replace the main noun subject with a more frequent noun subject that is likely to appear in the context of the original rare concept.
  c. Generate a text sequence that starts from the text with replaced frequent concepts and ends with the text with the original rare concepts.
  d. Additionally, please provide how much local visual detail is needed to draw the rare concept on a scale of 1 (minimal detail needed) to 5 (local detail essential), and explain why. Please give the score according to the degree of locality used to draw the visual concept.
The output should follow the format of the examples below:

**\<In-context Examples\>**
Input: {USER PROMPT}
Output:

Table 10: In-context examples of the LLM prompt for R2F.

Input: A hairy hamburger
Output:
  a. Rare Concept: A hairy hamburger
  b. Main Noun Subject: hamburger, Context: hairy, Replaced Frequent Subject: object
  c. A hairy object BREAK A hairy hamburger
  d. Extremely local visual details are needed to draw the hairs. Visual Detail Level: 5
  Visual Detail Level: 5
  Final Prompt Sequence: A hairy object BREAK A hairy hamburger

Input: A banana-shaped apple
Output:
  a. Rare concept: A banana-shaped apple
  b. Main noun object: apple, Context: banana-shaped, Replaced frequent object: red object
  c. A banana-shaped red object BREAK A banana-shaped apple
  d. Minimal local details are needed to draw the rough shape of banana. Visual Detail Level: 1
  Visual Detail Level: 1
  Final Prompt Sequence: A banana-shaped red object BREAK A banana-shaped apple

Input: A running dog
Output:
  No rare concept was found in the given input text.
  Visual Detail Level: 0
  Final Prompt Sequence: A running dog

Input: A horned lion and a hairy frog
Output:
  a. Rare concept: A horned lion
  b. Main noun subject: lion, Context: horned, Replaced frequent subject: animal
  c. A horned animal BREAK A horned lion
  d. Moderate local visual details are needed to draw horns. Visual Detail Level: 3
  AND
  a. Rare concept: A hairy frog
  b. Main noun subject: frog, Context: a hairy, Replaced frequent subject: animal
  c. A hairy animal BREAK A hairy frog
  d. Extremely local visual details are needed to draw hairs. Visual Detail Level: 5
  Visual Detail Level: 3 AND 5
  Final Prompt Sequence: A horned animal BREAK A horned lion AND A hairy animal BREAK A hairy lion

## C  LLM INSTRUCTION AND ALGORITHM PSEUDOCODE FOR R2F+

Table 11 details the complete LLM prompt for R2F+. Since the generation process of R2F+ requires detailed information, including rare-to-frequent concept mappings and region guides, we configure the LLM prompt to generate output in a structured JSON format.

Algorithm 1 describes the pseudocode for R2F+. As discussed in Section 3.3, the algorithm consists of three stages: (1) region-aware rare-to-frequent concept mapping, (2) masked latent generation via object-wise R2F, and (3) region-controlled alternating concept guidance.

Table 11: Full LLM prompt for R2F+ to generate region-aware rare-to-frequent concept mappings.

**<System Prompt>**
You are a helper language model for a text-to-image generation program that aims to create images based on input text.
Given the prompt describing the image, your task is to generate a stringified json object containing details.
Print a stringified JSON object instead of a code block.
**<User Prompt>**
Let's think step by step.

STEP 1. Identify the objects from the original prompt, and assign each object a key in the form of #1, #2, etc.
If some object appears multiple times in the prompt, assign a different key each time it appears.
For instance, in the case of "three X", each 'X' should be assigned different keys.
The closer the object is to the front, the higher its number should be.
Examples for STEP 1: **<In-Context Examples for STEP 1>**

STEP 2. Generate a base prompt, which each object is substituted by its key.
If an object appears multiple times, list each occurrence and separate them with "and".
For instance, if there are "three X" and each 'X' has keys '#2', '#3', and '#4', it should be written as '#2 and #3 and #4'.
The base prompt should contain exact details which the original prompt has.
Examples for STEP 2: **<In-Context Examples for STEP 2>**

STEP 3. For each object, generate a prompt that can be used to generate that specific object.
The object prompt should have exactly one placeholder of form '#N', which is the key of the target object.
Examples for STEP 3: **<In-Context Examples for STEP 3>**

STEP 4. Generate a bounding box (bbox) for each object.
The bounding box is a list of four numbers denoting [top-left x coordinate, top-left y coordinate, botom-right x coordinate, bottom-right y coordinate]
Each number is a real value between 0 and 1. The top-left coordinate of the image is (0, 0), and the bottom-right coordinate is (1, 1).
The bounding box should not go beyond the boundaries of the image.
Generate a bounding box considering the relationships between objects. The overall image should be balanced and centered.
Determine the width and height of the bounding box considering the shape of the object.
If two objects can be seperated, their bounding boxes should not overlap. Make a gap between their bounding boxes.
Also, try avoid too small (width or height less than 0.2), too narrow, or too wide bounding boxes.
Examples for STEP 4: **<In-Context Examples for STEP 4>**

STEP 5. Identify rare concepts from each object, and find relevant frequent concepts.
The program often struggles to accurately generate images when the input text contains rare concepts that are not commonly found in reality.
To address this, when a rare concept is identified in the input text, you should replace it with relevant yet more frequent concepts.
The replaced frequent concepts may include parent concepts and umbrella terms.
This will help the text-to-image generation program produce better-aligned images.
You can perform the following process step by step:
a. Identify and extract any rare concepts from the provided input text.
b. Replace the extracted rare concept with a more frequent concept. Specifically, split the rare concept into the main noun subject and the context, and replace the main noun subject with a more frequent noun subject that is likely to appear in the context of the original rare concept. Ensure that the replaced frequent noun subject retains the properties of the original main noun subject as much as possible while being appropriate to the context of the rare concept. If necessary, you may use multiple frequent concepts step by step to narrow down from the general to the specific. For example, object -> animal -> crocodile. Try to keep the number of frequent concepts small. Usually, one frequent concept is enough.
c. Generate a text sequence that starts from the text with replaced frequent concepts and ends with the text with the original rare concepts. If there are multiple frequent concepts, order them from general to specific.
Examples for STEP 5: **<In-Context Examples for STEP 5>**

STEP 6. Assign visual detail level to each frequent concept.
A visual detail level denotes how much local visual detail is needed to draw the rare concept on a scale of 1 to 5.
The list of visual detail levels should be increasing.
Examples for STEP 6: **<In-Context Examples for STEP 6>**

STEP 7. Organize the information into a single JSON object.
The JSON object should be in the following form.

```
{
    "original_prompt": str,                         // The original prompt
    "base_prompt": str,                             // The base prompt generated in STEP 2
    "objects": {                                    // The objects appearing in the image
        "#1": {                                     // The first object
            "prompt": str,                          // The object prompt generated in STEP 3
            "object": str,                          // The object found in STEP 1
            "r2f": list[str],                       // The freqent concepts generated in STEP 5
            "visual_detail_levels": list[int],      // Visual detail levels assigned in STEP 6
            "bbox": [float, float, float, float],   // The bbox generated in STEP 4
        },
        "#2": {                                     // The second object
            ...
        },
        ...
    }
}
```

Examples for STEP 7: **<In-Context Examples for STEP 7>**

STEP 8. Stringify the JSON object.
Examples for STEP 8: **<In-Context Examples for STEP 8>**

Input: "{INPUT}"
Output:

---

**Algorithm 1** Algorithm for R2F+

---

INPUT: $\mathbf{c}_{\text{input}}$: an input prompt,
    $T_{\text{CA}}$: the number of steps for performing cross-attention control,
    $N_{\text{CA}}$: the number of iterations for performing gradient descent in each cross-attention control step,
    $\eta_{\text{CA}}$: the coefficient for cross-attention control,
    $T_{\text{LF}}$: the number of steps for performing latent fusion
OUTPUT: $\mathbf{x}_0$: an overall image

1:   /* Stage 1. Region-aware Rare-to-frequent Concept Mapping */
2:   $\left(\mathbf{c}_{\text{base}}, \{(\mathbf{c}^i, \mathbf{c}_R^i, \mathbf{c}_F^i, V^i, R^i)\}_{i=1}^m\right) \leftarrow$ GETRESPONSEFROMLLM$(\mathbf{c}_{\text{input}})$
3:   /* Stage 2. Masked Latent Generation via Object-wise R2F */
4:   **for** $i = 1$ **to** $m$ **do**
5:       $\bar{\mathbf{z}}_T^i \sim \mathcal{N}(0, I)$
6:       **for** $t = T$ **to** $1$ **do**
7:           $\tilde{\mathbf{c}}_t^i \leftarrow$ GETSCHEDULEDPROMPT$(t, \mathbf{c}^i, \{(\mathbf{c}_R^i, \mathbf{c}_F^i, V^i)\})$
8:           /* Optional: perform additional cross-attention control */
9:           **if** $t > T - T_{\text{CA}}$ **do**
10:               $\bar{\mathbf{z}}_t^i \leftarrow$ CROSSATTENTIONCONTROL$(\bar{\mathbf{z}}_t^i, \{(\tilde{\mathbf{c}}_t^i, R^i)\})$
11:           $\bar{\mathbf{z}}_{t-1}^i \leftarrow p_\theta(\bar{\mathbf{z}}_t^i, \mathbf{c}_t^i)$
12:       $M^i \leftarrow$ GETMASK$(\text{VAE}(\bar{\mathbf{z}}_0^i), \mathbf{c}_R^i)$
13:       $\bar{\mathbf{z}}_{0:T}^i \leftarrow$ REFINELATENTS$(\bar{\mathbf{z}}_{0:T}^i, R^i)$
14:       $M^i \leftarrow$ REFINEMASK$(M^i, R^i)$
15:   /* Stage 3. Region-controlled Alternating Concept Guidance */
16:   $\mathbf{z}_T \sim \mathcal{N}(0, I)$
17:   **for** $t = T$ **to** $1$ **do**
18:       $\tilde{\mathbf{c}}_t \leftarrow$ GETSCHEDULEDPROMPT$(t, \mathbf{c}_{\text{base}}, \{(\mathbf{c}_R^i, \mathbf{c}_F^i, V^i)\}_{i=1}^m)$
19:       **if** $t > T - T_{\text{CA}}$ **do**
20:           $\mathbf{z}_t \leftarrow$ CROSSATTENTIONCONTROL$(\mathbf{z}_t, \{(\bar{\mathbf{c}}_t^i, R^i)\}_{i=1}^m)$
21:       $\mathbf{z}_{t-1} \leftarrow p_\theta(\mathbf{z}_t, \tilde{\mathbf{c}}_t)$
22:       **if** $t > T - T_{\text{LF}}$ **do**
23:           $\mathbf{z}_{t-1} \leftarrow$ LATENTFUSION$(\mathbf{z}_{t-1}, \{(\bar{\mathbf{z}}_{t-1}^i, M^i)\}_{i=1}^m)$
24:   $\mathbf{x}_0 \leftarrow$ VAE$(\mathbf{z}_0)$
25:   **return** $\mathbf{x}_0$;

---

26:   **function** GETSCHEDULEDPROMPT $\left(t, \mathbf{c}, \{(\mathbf{c}_R^i, \mathbf{c}_F^i, V^i)\}_{i=1}^k\right)$
27:       **for** $i = 1$ **to** $k$ **do**
28:           Determine the stop point $S_i$ from the visual detail level $V^i$
29:           **if** $t > S_i$ **and** $t\%2 = 1$ **do**
30:               In the prompt $\mathbf{c}$, find the rare concept $\mathbf{c}_R^i$ and replace it with the frequent concept $\mathbf{c}_F^i$
31:       **return** $\mathbf{c}$;

---

32:   **function** CROSSATTENTIONCONTROL $\left(\mathbf{z}_t, \{(\mathbf{c}^i, R^i)\}_{i=1}^k\right)$
33:       **for** $j = 1$ **to** $N_{\text{CA}}$ **do**
34:           **for** $i = 1$ **to** $k$ **do**
35:               Obtain cross-attention map $A^i$ from the cross-attention layers of the diffusion step $p_\theta(\mathbf{z}_t, \mathbf{c}^i)$
36:               Calculate $E(A^i, R^i)$, where

$$E(A, R) = \left(1 - \frac{\sum_{(x,y) \in R} A_{x,y}}{\sum_{(x,y)} A_{x,y}}\right)^2$$

37:       $\mathbf{z}_t \leftarrow \mathbf{z}_t - \eta_{\text{CA}} \nabla_{\mathbf{z}_t} \sum_{i=1}^k E(A^i, R^i)$
38:       **return** $\mathbf{z}_t$;

---

39:   **function** LATENTFUSION $\left(\mathbf{z}, \{(\bar{\mathbf{z}}^i, M^i)\}_{i=1}^k\right)$
40:       **for** $i = 1$ **to** $k$ **do**
41:           $\mathbf{z} \leftarrow \mathbf{z} \odot (1 - M^i) + \bar{\mathbf{z}}^i \odot M^i$
42:       **return** $\mathbf{z}$;

---

In the first stage, LLM decomposes the given input prompt $\mathbf{c}_{\text{input}}$ into sub-prompts $\{\mathbf{c}^i\}_{i=1}^m$ and generates region-aware rare-to-frequent concept mapping $\{(\mathbf{c}^i, \mathbf{c}_R^i, \mathbf{c}_F^i, V^i, R^i)\}_{i=1}^m$. In addition, the overall base prompt $\mathbf{c}_{\text{base}}$ is also generated to guide the overall image generation (line 2).

During the second stage, we process each object $i$ individually to generate the object-wise latents $\bar{\mathbf{z}}_{T:0}^i$ and the mask $M^i$ (line 4). For each diffusion step $t$, the next latent $\bar{\mathbf{z}}_{t-1}^i$ is sampled with guidance from the scheduled prompt $\tilde{\mathbf{c}}_t^i$ (line 11). Similar to R2F, the scheduled prompt $\tilde{\mathbf{c}}_t^i$ is determined by the rare concept $\mathbf{c}_R^i$, the frequent concept $\mathbf{c}_F^i$, and the visual detail level $V^i$ (line 7). After obtaining the final latent $\bar{\mathbf{z}}_0^i$, the mask $M^i$ is generated (line 12). To ensure alignment with the region guide $R^i$, the latents $\bar{\mathbf{z}}_{0:T}^i$ and the mask $M^i$ go through a refinement process (lines 13-14), which involves resizing and shifting.

To obtain the object mask $M^i$ for object $i$ from its object-wise image, we follow a two-step process. First, an object detection model such as Grounding DINO (Liu et al., 2023b) identifies the bounding

box corresponding to $\mathbf{c}_R^i$. This bounding box may differ from the provided region guide $R^i$. Next, a segmentation model such as SAM (Kirillov et al., 2023) generates the segmentation mask based on the bounding box. The resulting mask is resized from the image space to the latent space.

In the final stage, the overall image is generated by region-controlled concept guidance. Similar to the second stage, the scheduled prompt $\tilde{\mathbf{c}}_t$ is determined by the rare concept $\mathbf{c}_R^i$, the frequent concept $\mathbf{c}_F^i$, and the visual detail level $V^i$ for each image $i$ (line 18). Prior to the diffusion step (line 21), the cross-attention control is applied to align the latent with region guides (lines 19-20). Following the diffusion step, the latent fusion is performed to integrate the object-wise masked latents into the overall latent (lines 22-23).

Optionally, during the second stage of the algorithm, additional cross-attention control can be applied to ensure that the object-wise latents align with their region guides before the refinement process (lines 9-10). With this method, resizing latents can be avoided during the refinement process (lines 13-14). We adopt this method in our practical implementation. For better results, we reposition the bounding box of each object region to the center of the image, generate object-wise images with these centered regions, and then shift each masked object latent back to its original position, as demonstrated in Lian et al. (2023).

In our implementation, to accurately identify multiple rare concepts from prompts and replace them with frequent concepts, LLM substitutes each object in the base prompt $\mathbf{c}_{\text{base}}$ and the object-wise sub-prompts $\mathbf{c}_i$ with a special key of the form "#N". These keys are then replaced with corresponding rare or frequent concepts during the prompt scheduling.

In R2F+, four new hyperparameters are introduced. $T_{\text{CA}}$ represents the number of steps for performing cross-attention control. $N_{\text{CA}}$ represents the number of iterations for performing gradient descent in each cross-attention control step. $\eta_{\text{CA}}$ represents the coefficient for cross-attention control, and $T_{\text{LF}}$ represents the number of steps for performing latent fusion. In our experiment, we used $T_{\text{CA}} = 10$, $N_{\text{CA}} = 5$, $\eta_{\text{CA}} = 30$, and $T_{\text{LF}} = 20$.

## D  DETAILS FOR CONSTRUCTING RAREBENCH

**Overview.** RareBench aims to evaluate the T2I model's compositional generation ability for rare concept prompts across single- and multi-objects. For single-object prompts, we categorize visual concept attributes in *five* cases: (1) property, (2) shape, (3) texture, (4) action, and (5) complex, which is a mixture of the attributes. For multi-object prompts, we categorize the way to combine multiple prompts/concepts with single-object in *three* cases: (1) concatenation, (2) relation, and (3) complex. To ensure the prompt rareness, we construct the prompt set in a two-stage process: (i) *Rare concept composition generation by GPT-4o*, and (ii) *Rare prompt selection by human*.

Table 12: Attribute categories and examples in RareBench. These attributes are combined with contextually rarely appeared objects to compose prompts.

| Property | hairy, horned, wooly, bearded, mustachioed, thorny, spiky, wrinkled, spotted, wigged, hairless |
|---|---|
| Shape | banana-shaped, star-shaped, ax-shaped, butterfly-shaped, oval-shaped, donut-shaped, hand-shaped, gear-shaped, heart-shaped, diamond-shaped |
| Texture | flower-patterned, zebra-striped, tiger-striped, black-white-checkered, made of marble, made of diamonds, made of plastic, made of glass, made of steel, made of cloud |
| Action | dancing, walking, running, crawling, flying, swimming, driving a car, yawning, smiling, crying, cheerleading |

**Rare concept candidate generation by GPT-4o.** For each attribute category, we first prepare attribute examples that can be used to compose prompts. Table 12 shows these attributes examples. Based on this list, we ask GPT-4o to retrieve a candidate set of objects that rarely co-exist in reality. Specifically, we prompt 'Given an attribute EXAMPLE, generate a list of objects that are rarely co-exist or almost impossible to appear in reality. Also, please generate a short proper caption consisting of the attribute and object.'. Then, for each attribute example of each category, we obtain rare concept candidate prompts. For multi-object cases with relation case, we expose GPT-4o multiple prompt

with a single object, then ask to generate a prompt with multi-objects consisting of the combination of the exposed prompts by adding a proper relation to combine two prompts.

**Rare concept selection by human.** To further make the prompt set rarer, we involve human labor to pick more rare and interesting prompt examples to generate by T2I models. For the two complex cases, we get the rare and interesting prompt set directly from humans to make our benchmark to more precisely investigate real use cases of T2I models.

## E   GPT INSTRUCTION FOR CALCULATING % RARENESS

Since real-world T2I datasets typically exhibit a *long-tailed* nature (Xu et al., 2023; Park et al., 2020), they contain many rare concepts (in the tail distributions). To measure whether a prompt contains rare concepts that are very difficult to observe in the real-world, we ask GPT4 with the yes or no binary question using the following instructions. *"You are an assistant to evaluate if the text prompt contains rare concepts that exist infrequently or not in the real world. Evaluate if rare concepts are contained in the text prompt: PROMPT, The answer format should be YES or NO, without any reasoning.".* Formally, the % rareness of each test dataset $\mathcal{C}_{\text{test}}$ is calculated as $\%\text{Rareness}(\mathcal{C}_{\text{test}}) = 1/|\mathcal{C}_{\text{test}}| \sum_{c \in \mathcal{C}_{\text{test}}} \mathbb{1}(\text{GPT}_{\text{rare}}(c) == \text{Yes})$, where $\text{GPT}_{\text{rare}}(c)$ is the binary answer of rareness from GPT.

## F   DETAILS FOR EVALUATION

**GPT-based Evaluation.** We leverage GPT-4o to evaluate the image-text alignment between the prompt and the generated image. The evaluation is based on a scoring scale from 1 to 5, where a score of 5 represents a perfect match between the text and the image, and a score of 1 indicates that the generated image completely fails to capture any aspect of the given prompt. Table 13 presents the complete prompt with a full scoring rubric. We convert the original score scale $\{1, 2, 3, 4, 5\}$ to $\{0, 25, 50, 75, 100\}$, which is reflected in the reported results.

**Human Evaluation.** We collect human evaluation scores from *ten* different participants. For each prompt, all the generated images produced by R2F and baseline approaches are presented simultaneously to each participant. This comparative evaluation allows participants to provide more accurate evaluations by assessing all methods side by side (Sun et al., 2023). During evaluation, the name of all the methods is fully anonymized, and their order is randomly shuffled for each prompt to prevent any bias. Participants follow the same scoring criteria used in the GPT-based evaluation. Score scale is also converted from $\{1, 2, 3, 4, 5\}$ to $\{0, 25, 50, 75, 100\}$.

Table 13: Full LLM instruction for evaluation.

You are my assistant to evaluate the correspondence of the image to a given text prompt.
Focus on the objects in the image and their attributes (such as color, shape, texture), spatial layout, and action relationships. According to the image and your previous answer, evaluate how well the image aligns with the text prompt: **[PROMPT]**

Give a score from 0 to 5, according to the criteria:
5: image perfectly matches the content of the text prompt, with no discrepancies.
4: image portrayed most of the content of the text prompt but with minor discrepancies.
3: image depicted some elements in the text prompt, but ignored some key parts or details.
2: image depicted few elements in the text prompt, and ignored many key parts or details.
1: image failed to convey the full scope in the text prompt.

Provide your score and explanation (within 20 words) in the following format:
### SCORE: score
### EXPLANATION: explanation

## G   FURTHER VISUALIZATION

Figure 13 shows *uncurated* generated images of R2F on RareBench. We randomly select 8 prompts from RareBench and generate images with 8 random seeds. Overall, most of the generated images are well-aligned with the input prompt, without compromising the naturalness and quality.

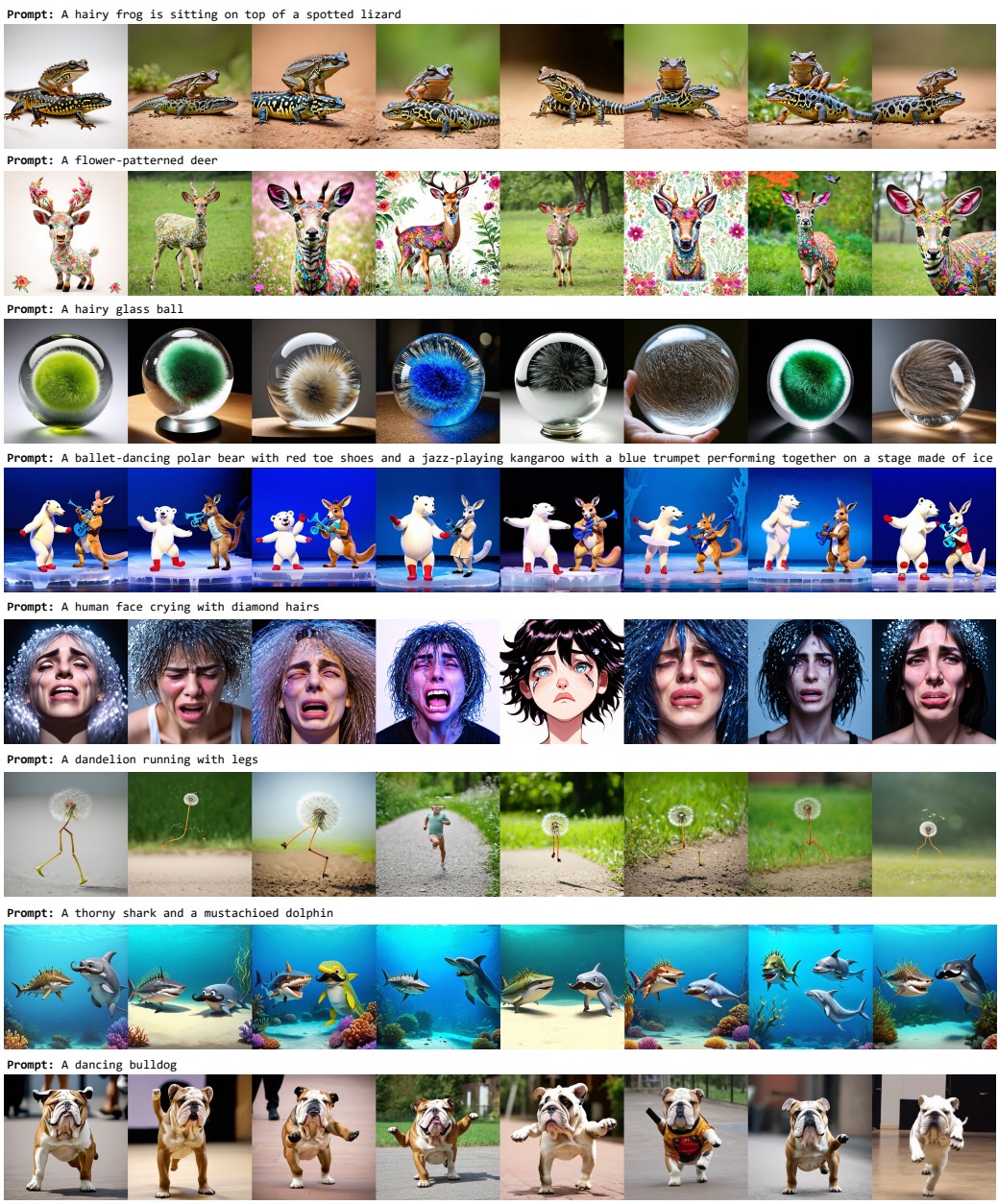

Figure 13: *Uncurated* visualization results of R2F on RareBench. Images are generated from 8 random prompts with 8 random seeds.

## H    EXAMPLES FOR RARE-TO-FREQUENT CONCEPT MAPPING

Table 14 shows examples for R2F concept mapping generated by GPT-4o. All examples are selected from RareBench. By leveraging the state-of-the-art LLM, R2F successfully split the prompt by objects, identify rare concepts, extract their relevant yet more frequent concepts, and their visual detail level to generate images.

## I    FAILURE RESULTS OF T2I-COMPBENCH'S AUTO-EVALUATION METRICS

Figure 14 shows the failure results of auto-evaluation metrics in T2I-CompBench. In the left case, while both SD3.0 and R2F generate appropriate images well-following the input prompt '*A big hippopotamus and a small mouse*', R2F got very low BLIP score 0.0322. Also, in the right case,

Table 14: Examples for LLM-generated R2F concept mapping.

| Original Prompt | Rare-to-frequent Concept Mapping | | | |
| | Sub-prompt | Rare concept | Frequent concept | Visual detail level |
| --- | --- | --- | --- | --- |
| A hairless sheep | A hairless sheep | A hairless sheep | A hairless animal | 3 |
| A donut shaped earth | A donut shaped earth | A donut shaped earth | A donut shaped blue object | 1 |
| A zebra striped duck | A zebra striped duck | A zebra striped duck | A zebra striped animal | 3 |
| A cheetah driving a car | A cheetah driving a car | A cheetah driving a car | An animal driving a car | 2 |
| A polar bear with wings of a phoenix | A polar bear with wings of a phoenix | A polar bear with wings of a phoenix | a creature with bird wings | 4 |
| a cactus made of steel and two sunflowers made of glass | a cactus made of steel | a cactus made of steel | a spiky object made of steel | 4 |
| | two sunflowers made of glass | two sunflowers made of glass | two yellow items made of glass | 3 |
| A mustachioed strawberry driving a banana shaped car is following a dancing koala | A mustachioed strawberry | A mustachioed strawberry | a mustachioed fruit | 4 |
| | a banana shaped car | a banana shaped car | a banana shaped object | 1 |
| | a dancing koala | a dancing koala | a dancing animal | 2 |
| A rabbit in medieval armor and a raccoon in pajamas milling on the moon | A rabbit in medieval armor | A rabbit in medieval armor | an animal in medieval armor | 3 |
| | a raccoon in pajamas | a raccoon in pajamas | an animal in pajamas | 4 |
| A knighted turtle and a wizarding owl debating philosophy in an underwater library made of coral | A knighted turtle | A knighted turtle | a decorated animal | 3 |
| | a wizarding owl | a wizarding owl | a magical bird | 3 |
| | an underwater library made of coral | an underwater library made of coral | an underwater structure made of coral | 4 |

**SD3.0)** BLIP score: 0.9307  **R2F)** BLIP score: 0.0322  **SD3.0)** BLIP score: 0.9482  **R2F)** BLIP score: 0.9520

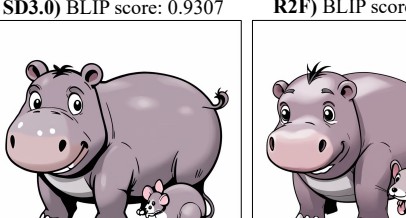

**Prompt:** A big hippopotamus and a small mouse

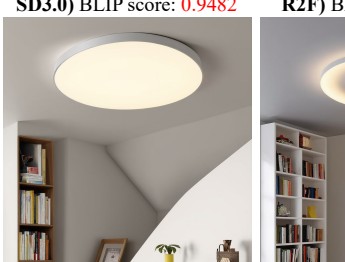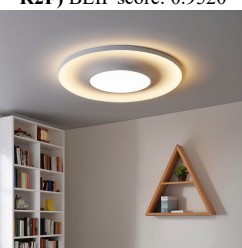

**Prompt:** A circular ceiling light and a triangular bookshelf

Figure 14: Failure results of T2I-Compbench's auto-evaluation metrics. BLIP score is a T2I alignment score calculated by using an open-source multi-modal model, BLIP (Li et al., 2022), which is often more inaccurate than a proprietary multi-modal LLM, GPT-4o.

while SD3.0 fails to generate '*a triangular bookshelf*', it got a very high BLIP score $0.9482$ which is similar to that of R2F, $0.9520$. This inaccuracy may be the reason why the auto-eval metrics in T2I-CompBench are not well-aligned to the T2I alignment score from GPT-4o, which is well-aligned with the human evaluation.

## J   MORE VISUALIZATION RESULTS OF R2F+

Figure 15 shows more visualization examples and failure cases of R2F+ on RareBench. As shown in the first row (i.e., Successful Cases), R2F+ produces highly controllable image generation results when the bounding boxes are non-overlapped and assigned with proper sizes. However, as shown in the second row (i.e., Failure Cases), when the bounding boxes are overlapped or too small, which is usually produced by LLM when the prompt is long and complex, it fails to accurately generate

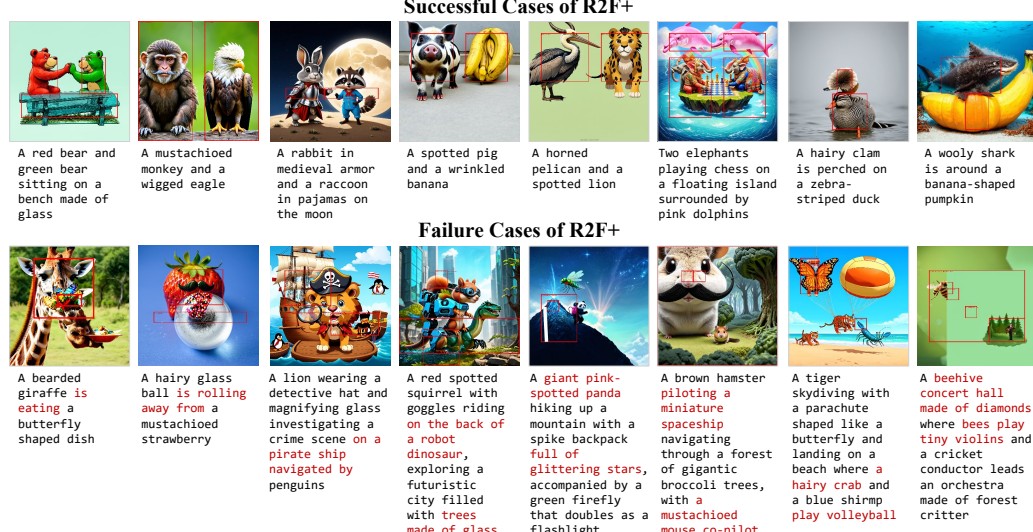

Figure 15: More visualization examples with successful and failure cases of R2F+.

visual concepts. For example, the boundaries between objects are blurred, and the detailed attributes disappear, which is often reported by the existing literature of region-guided diffusion (Chen et al., 2024c; Yang et al., 2024).

## K ACCELERATED R2F USING FLUX WITH 4-STEP INFERENCE.

Here, we further propose an accelerated version of R2F integrated with FLUX-schnell which only requires 4 or fewer steps to generate each image. In scenarios with short inference lengths, the efficacy of alternating guidance may diminish. Therefore, while maintaining R2F's main idea of exposing frequent concepts to the diffusion sampling process, we propose the Composable method (Same as the one we used in Section 4.4) as an alternative.

**Configuration.** Given a pair of rare-frequent concept prompts, Composable blends the text embeddings of two prompts and uses them as the input for diffusion sampling steps. For the guidance length of 4 steps, we adopted its interpolation configuration as follows: (1) For concepts with a visual detail level from 1 to 3, we applied the composable method only for the first step with the blending factor $\alpha$ of 0.3. (2) For concepts with a visual detail level from 4 to 5, we further applied it until the second step with the decreased blending factor $\alpha$ of 0.3. Only the original rare prompt was exposed in the final third and fourth steps.

**Results.** Table 15 shows the 4-step inference result of R2F combined with FLUX-schnell on RareBench single case. Owing to the adopted frequent concept exposure approach based on the Composable method, R2F improves the compositional generation ability of FLUX even with short guidance steps of 4. Figure 16

Table 15: 4-step inference result of R2F combined with FLUX-schnell.

| RareBench | Property | Shape | Texture | Action | Complex |
|---|---|---|---|---|---|
| FLUX | 72.5 | 68.1 | 49.3 | 61.2 | **73.7** |
| R2F$_{flux}$ | **78.7** | **75.0** | **56.8** | **67.5** | 68.7 |

visualizes the generated images of R2F combined with FLUX. Overall, R2F significantly enhances the T2I alignment of the generated image without compromising the image quality. This indicates the frequent concept exposure idea of R2F can be generalized to the latest acceleration methods with short sampling steps, which leads to a broader impact on many real-world applications where fast inference time is crucial.

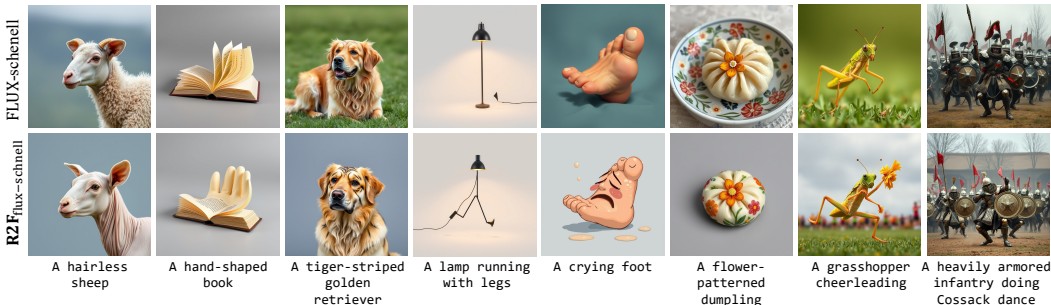

Figure 16: Generated images of R2F using FLUX as the backbone with a 4-step denoising process.

## L  VISUALIZATION RESULTS OF R2F WHEN EXPOSING FREQUENT CONCEPT UNTIL THE LAST STEP.

Figure 17 shows the generated images of R2F when using concept alternating at the beginning steps and using frequent concepts at the last steps. Overall, the generated image tends to align more closely with the frequent prompt rather than the original rare prompt. This is because diffusion is a step-wise denoising process where the generated image depends more on the prompt that has been used lately.

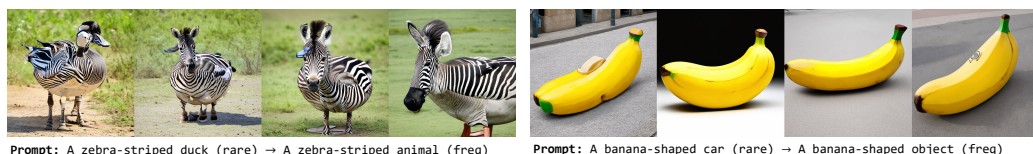

**Prompt:** A zebra-striped duck (rare) → A zebra-striped animal (freq)     **Prompt:** A banana-shaped car (rare) → A banana-shaped object (freq)

Figure 17: Visualization examples of R2F when exposing frequent concept until the last step.

## M  EFFECT OF LAION-400M DATASET FOR FINDING RARE-FREQUENT CONCEPT MAPPING.

Here, we investigate the effect of LAION-400M dataset (Schuhmann et al., 2022), one of the largest open-source image-caption datasets available online, to enhance the performance of R2F in terms of finding more appropriate rare-frequent concept mapping.

**Setup.** Given a prompt (e.g., "*a bearded apple*"), if it contains a rare attribute word ("*bearded*"), we measure the frequency of its *next* words in the LAION-400M. For example, if there are 100 captions containing "*bearded man*" in LAION-400M, the frequency of "*man*" for the attribute "*bearded*" is calculated as 100. We then integrated these next-word frequencies into the R2F process in two ways: **(1) using the most frequent next word as the frequent concept**. For each rare attribute word, we extract the most frequent subsequent noun from LAION and directly use it for the noun of the frequent concept. For example, if "*man*" appears most frequently after "*bearded*", we use "*bearded man*" as the frequent concept for the original rare concept such as "*bearded apple*". **(2) providing Top20 next word frequency in LLM prompt**. In this case, we extract the top 20 frequent next words from LAION-400M, and provide them to the LLM prompting for identifying rare-to-frequent concept mapping with the following instruction; "...*When finding frequent concepts for extracted rare concepts, please consider the words that appeared most frequently after the attribute word of the rare concept in the LAION image caption dataset. The list of the top 20 words is as follows and is in the format of ('next word', 'count'). EXAMPLES...*".

**Result.** Table 16 shows the effect of LAION-400M for finding rare-frequent concept mapping. While we expose the word frequency information from LAION-400M into R2F, the performance of these variants is not higher than the original R2F. This may be because the LAION captions are low-quality (e.g., captions are mostly alt texts crawled from the web) and recent

Table 16: Effect of LAION-400M for finding rare-frequent concept mapping in R2F.

| Models | SD3 | R2F | R2F w/ (1) | R2F w/ (2) |
|---|---|---|---|---|
| RareBench$_{property}$ | 49.4 | 89.4 | 81.3 | 85.9 |

models such as SD3.0 are trained on more high-quality undisclosed image-caption datasets (Betker et al., 2023), which potentially diverges from the distribution of LAION captions. For example, the top 5 subsequent words following "bearded" in LAION-400M are ('man', 8772), ('dragon', 5996), ('collie', 3573), ('iris', 2153), and ('dragons', 1087), showing a discrepancy with common sense knowledge. Nevertheless, all the variants outperform SD3.0, which indicates the effectiveness of the fundamental idea of R2F's frequent concept exposure.

## N    QUANTITATIVE IMAGE QUALITY ANALYSIS OF R2F+.

**Metrics**. For quantitative image quality analysis of R2F+, we use *three* popular image quality scores, including LAION-aesthetic (Schuhmann et al., 2022), PickScore (Kirstain et al., 2023), and ImageReward (Xu et al., 2024).

**Result**. Table 17 compares the image quality scores of R2F+ with R2F. Overall, there is no significant difference between the two models in terms of image quality scores, indicating that R2F+ can achieve better spatial composition without much comprising the image quality compared to R2F.

Table 17: Aesthetic scores of generated images by R2F+ on RareBench multi-object case.

| Scores | LAION-aesthetic | PickScore | ImageReward |
|---|---|---|---|
| R2F | 3.980±0.361 | 0.226±0.009 | 0.626±0.029 |
| R2F+ | 3.887±0.353 | 0.222±0.009 | 0.609±0.033 |

## O    GPU TIME AND MEMORY ANALYSIS.

To further show the practicality of our algorithms, we provide the GPU time and memory analysis on a single NVIDIA 40GB A100 GPU. We measure the GPU time and memory required to generate a prompt "*a horned lion and a wigged elephant*", which consists of two rare concepts "*horned lion*" and "*wigged elephant*". The results are shown in Table 18 (only time and memory taken for diffusion sampling steps are presented).

Table 18: GPU time and memory required to generate an image of prompt "a horned lion and a wigged elephant".

| Models | SD3 | R2F | R2F+ |
|---|---|---|---|
| Peak Memory (GB) | 31.52 | 31.76 | 35.08 |
| GPU Time (sec) | 20.04 | 20.60 | 72.37 |

**Peak Memory.** While R2F+ involves several latent and gradient computations, there is no significant difference in peak memory compared to R2F since it follows a sequential process. R2F requires approximately 31GB of peak memory, and R2F+ requires approximately 35GB of peak memory where an additional 4GB mostly comes from the gradient computations in cross-attention control.

**GPU Time.** The time taken for R2F+ is approximately 72 sec, which can be decomposed as 1) masked latent generation via object-wise R2F, which takes around 42 sec, and 2) region-controlled concept guidance, which takes around 30 sec. Specifically, for the process of 1), the generation of each object takes around 20 sec (same as R2F) with an additional 1 sec for masking, resulting in a total of 42 sec for two objects. For the process of 2), the majority of the increased computation time compared to R2F is attributed to attention control, which adds around 10 sec. Consequently, for a prompt with $N$ objects, the time complexity of R2F+ is expected as $N * (T + 1) + (T + 10)$, where $T$ is the inference time of R2F or SD3.

## P    BROAD APPLICATIONS.

Rare concept composition is essential in various applications that require the creation of *creative content*, such as designing characters and posters for comics, movies, and games. Creators in these domains should often produce content that has never existed, such as characters with elemental faces (e.g., fire or water), pirate ship-shaped spaceships, or trumpet-like guns. Therefore, rare concept composition could be considered a mainstream area for these creators.

Furthermore, the idea of frequent concept guidance can potentially be extended to *other modalities*, such as text-to-speech (TTS) (Liu et al., 2023a; Lyth & King, 2024; Lee et al., 2024) and text-to-music (Copet et al., 2024). For instance, TTS models have concept categories including speaker, intonation, and emotion. When generating speech such as "*an angry Trump speaking Catalan*", we might expose frequent concepts such as "*an angry Spanish speaking in Catalan*" to improve the composition performance of diffusion-based TTS models.

R2F+ is useful in many applications where the layout-aware composition is essential. (1) Layout-aware image/poster design. When a user creator wants to place an object in a specific position within an image for poster design, R2F is insufficient because it cannot adjust absolute positions. In such cases, R2F+ becomes essential. (2) Data synthesis for enhancing spatial understanding of foundation models. Recent multi-modal LLMs (e.g., LLaVA (Liu et al., 2024; Park et al., 2024)) and pre-trained VLMs (e.g., CLIP (Radford et al., 2021)) are known to exhibit weaknesses in spatial understanding (Chen et al., 2024a), so several studies have attempted to enhance their performance with spatiality-aware image synthesis (e.g., generating images that accurately captures spatial information in text prompts). R2F+ has the potential to enhance the performance of these foundation models by serving as a data synthesis method, when spatial composition is more critical than image quality.

