# OpenReview forum: "Rare-to-Frequent: Unlocking Compositional Generation Power of Diffusion Models on Rare Concepts with LLM Guidance"
_ICLR.cc/2025/Conference — ICLR 2025 Spotlight_

### Official Review · Reviewer_7ZtE · 2024-10-28

**Soundness:** 3
**Presentation:** 4
**Contribution:** 3
**Rating:** 8
**Confidence:** 4

**Summary:**

The paper introduces an innovative method for compositional generation of rare concepts. It demonstrates both theoretically and empirically that incorporating frequent concepts related to the target rare concepts leads to more accurate compositions. Building on this analysis, the Rare2Frequent (R2F) approach is presented, which strategically guides the transition from rare to frequent concepts during diffusion inference by utilizing the extensive semantic knowledge available in large language models (LLMs). R2F undergoes comprehensive evaluation, both qualitatively and quantitatively, achieving state-of-the-art results on multiple benchmarks, along with the introduction of a new benchmark for rare compositions.

**Strengths:**

- The paper is clearly written and well-organized.
- The results, both qualitative and quantitative, are impressive.
- Although the concept of transferring knowledge from frequent to rare concepts has been explored in the context of domain adaptation and long-tail learning [1,2,3], its application in diffusion models for image generation is novel.
- A significant new benchmark, RareBench, is introduced to assess the generation of rare concept compositions.
- The proposed approach is applied to various diffusion models (SD3.0, Flux, RPG, and region-guided diffusion), demonstrating its effectiveness.


[1] Parisot., et al. (2022) Long-tail Recognition via Compositional Knowledge Transfer.
[2] Samuel., et al. (2020) From Generalized zero-shot learning to long-tail with class descriptors.
[3] Jing., et al. (2021) Towards Fair Knowledge Transfer for Imbalanced Domain Adaptation.

**Weaknesses:**

My primary concern is that what is deemed rare for the diffusion model may not be considered rare for the LLM. Since the LLM lacks access to the training distribution of concepts used by the diffusion model, it may substitute rare concepts with other rare ones. Providing the LLM with the concept distribution from LION could enhance the results. This distribution has been published by [1].

[1] Samuel., et al. (2024) Generating images of rare concepts using pre-trained diffusion models. (https://github.com/dvirsamuel/SeedSelect/tree/main)

**Questions:**

See weaknesses.

---

> ### Author Response · Authors · 2024-11-19
> **Author's Response to Reviewer 7ZtE**
>
> We sincerely appreciate the reviewers' constructive comments and positive feedback on our manuscript.
>
> `W1. My primary concern is that what is deemed rare for the diffusion model may not be considered rare for the LLM. Since the LLM lacks access to the training distribution of concepts used by the diffusion model, it may substitute rare concepts with other rare ones. Providing the LLM with the concept distribution from LION could enhance the results. This distribution has been published by [1].`
>
> This is an excellent question. Because recent diffusion models have been trained on **billion-scale text-to-image datasets**, we could naturally expect that their distributions are closely aligned with LLMs. This may be the reason why our zero-shot LLM guidance performs well in our experiments. Meanwhile, following your suggestion, we provided captions from LAION-400M to assist LLMs in generating a rare-to-frequent concept mapping.
>
> **Setup.** Given a rare attribute word ("bearded") in a prompt ("a bearded apple"), we measured the frequency of all the subsequent words in the LAION dataset. For example, if there are 100 captions containing "bearded man" in LAION, the frequency of "man" for the attribute "bearded" is calculated as 100. We then integrated these next-word frequencies into the R2F process in two  ways:
> **(1) R2F with the most frequent subsequent word in LAION.** For each rare concept attribute word, we extract the most frequent subsequent noun from LAION and directly use it for the noun of the frequent concept. For example, if "man" appears most frequently after "bearded", we use "bearded man" as the frequent concept for the original rare concept such as "bearded giraffe".
> **(2) R2F using Top20 subsequent word frequency in LLM prompt.** In this case, we extract the top 20 frequent subsequent words from LAION, and add them to the LLM instruction for identifying rare-to-frequent concept mapping as follows:
>
> > ...When finding frequent concepts for extracted rare concepts, please consider the words that appeared most frequently after the attribute word of the rare concept in the LAION image caption dataset. The list of the top 20 words is as follows and is in the format of ('next word', 'count'). \n EXAMPLES...
>
>
> **Result.** The results for these R2F variants with LAION information are in the table below.
>
> | Models | SD3 | R2F | R2F with (1) | R2F with (2)|
> | ------------------ | ---- | ---- | ---- | ---- |
> | RareBench_property | 49.4 | 89.4 | 81.3 | 85.9 |
>
> While we expose the LAION information, the performance of these variants is not higher than our original R2F. This may be due to the low quality of LAION captions (e.g., captions are mostly alt texts that are crawled from the web), and because recent models such as SD3.0 are trained on more high-quality image-caption datasets [1], potentially diverging from the distribution of LAION captions. Indeed, the top 5 subsequent words following "bearded" in LAION were ('man', 8772), ('dragon', 5996), ('collie', 3573), ('iris', 2153), and ('dragons', 1087), showing a discrepancy with common sense knowledge.
>
> However, we believe that with access to high-quality diffusion training sets, the performance of LLM guidance can be further enhanced, and we will include this finding in the revised manuscript. Again, we greatly appreciate your insightful feedback.
>
> - **We added this analysis in Section M of the revised manuscript.**
>
> ---
> [1] Improving Image Generation with Better Captions, ArXiv, 2023

---

> > ### Comment · Reviewer_7ZtE · 2024-11-20
> >
> > Thanks for the clarification. I have read all the responses. I would like to keep my original score.

---

### Official Review · Reviewer_zqku · 2024-10-30

**Soundness:** 3
**Presentation:** 3
**Contribution:** 3
**Rating:** 6
**Confidence:** 4

**Summary:**

This paper deals with generating rare compositions of concepts, which is challenging for existing compositional generation methods.  The authors propose Rare-to-Frequent (R2F), which utilizes LLMs to plan and execute the overall rare-to-frequent concept guidance throughout the diffusion inference.  The paper improves R2F with the layout guidance to achieve more precise spatial-aware generation. Moreover, a new benchmark RareBench is proposed.

**Strengths:**

- This paper is well-written and easy to follow.
- The method is training-free. Experimental results show that R2F outperforms previous models on various metrics.
- It brings a new task to compositional generation or text-to-image generation.

**Weaknesses:**

- For applications, rare concept composition generation is still a relatively niche area, although I acknowledge that it is indeed a novel task within compositional generation. Have you considered exploring a broader range of application scenarios?
- For the computational cost, this paper adopts an approach similar to LMD to enhance R2F, resulting in R2F+, which involves substantial latent and gradient computations. A detailed comparison of computational and memory overhead with other methods is essential to assess the feasibility of the proposed approach.

**Questions:**

- I’m not entirely clear on the specific rules LLMs use to determine the “visual detail level.” In your writing, this measure is used in alternating concept guidance to set the guidance length for rare and frequent prompts, with more challenging rare concepts requiring extended guidance. However, LLMs lack knowledge of diffusion priors, which would inform the difficulty associated with generating certain objects or attributes.
- The example you give in Figure 4, where "plants made of glass", I don't think it is a frequent concept. Furthermore, in the initial stages of denoising, diffusion models primarily focus on generating rough visual features (e.g., shape, location). Consider the concept of “furry”; both “furry bird” and “furry tiger” are frequent concepts LLMs may output, yet there is a significant difference in the size and shape of these objects, which has a notable impact on the generated result. Thus, I question whether LLMs can reliably provide suitable frequent concepts.
- Is the design of R2F+ necessary？ In fact, layout-based methods have outstanding spatial awareness,  however, the trade-off is increased computational cost and a decline in image quality (in terms of detail, aesthetics, etc.). First, you need to conduct a comparative evaluation of R2F+ in terms of image quality. Additionally, as noted in Table 3’s T2I-CompBench, R2F achieves higher spatial metrics than both the layout-based method LMD and the LLM-based method RPG. Thus, expanding R2F to a layout-based approach may be unnecessary, as it would only improve spatial performance while significantly compromising image quality.
- You can consider using the IterComp[1], which is a backbone specifically designed for compositional generation and may lead to a more significant performance improvement.

I will revise my rating according to the author's feedback and the reviewer's discussion.

[1] IterComp: Iterative Composition-Aware Feedback Learning from Model Gallery for Text-to-Image Generation

**Details Of Ethics Concerns:**

This paper has no ethical concerns.

---

> ### Author Response · Authors · 2024-11-19
> **Author's Response to Reviewer zqku (1/3)**
>
> `W1. For applications, rare concept composition generation is still a relatively niche area, although I acknowledge that it is indeed a novel task within compositional generation. Have you considered exploring a broader range of application scenarios?`
>
>
> This is an excellent question. Rare concept composition is essential in various applications that require the creation of **creative content**, such as designing characters and posters for comics, movies, and games. Creators in these domains should often produce content that has never existed, such as characters with elemental faces (e.g., fire or water), pirate ship-shaped spaceships, or trumpet-like guns. Therefore, rare concept composition could be considered a mainstream area for these creators.
>
> Furthermore, our idea of frequent concept guidance can potentially be **extended to other modalities**, such as text-to-speech (TTS) [1,2,3] and text-to-music [4]. For instance, TTS models have concept categories including speaker, intonation, and emotion. When generating speech such as "an angry Trump speaking Catalan", we might expose frequent concepts such as "an angry Spanish speaking in Catalan" to improve the composition performance of diffusion-based TTS models.
>
> Thus, we believe that rare concept composition has a broader range of application scenarios.
>
> - **We included this discussion in Section P of the revised manuscript.**
>
> ---
>
> [1] AudioLDM: Text-to-Audio Generation with Latent Diffusion Models, ICML, 2023
>
> [2] Natural language guidance of high-fidelity text-to-speech with synthetic annotations, ArXiv, 2024
>
> [3] DiTTo-TTS: Efficient and Scalable Zero-Shot Text-to-Speech with Diffusion Transformer, ArXiv, 2024
>
> [4] Simple and Controllable Music Generation, NeurIPS, 2023
>
>
>
> `W2. For the computational cost, this paper adopts an approach similar to LMD to enhance R2F, resulting in R2F+, which involves substantial latent and gradient computations. A detailed comparison of computational and memory overhead with other methods is essential to assess the feasibility of the proposed approach.`
>
> We thank the reviewer for helping us improve our paper. The table below compares the A100 GPU time and memory required to generate a prompt "*a horned lion and a wigged elephant*", which consists of *two* rare concepts "*horned lion*" and "*wigged elephant*".
>
> | Models           | SD3   | R2F   | R2F+  |
> | ---------------- | ----- | ----- | ----- |
> | Peak Memory (GB) | 31.52 | 31.76 | 35.08 |
> | GPU Time (sec)   | 20.04 | 20.60 | 72.37 |
>
>
> **Peak memory.** While R2F+ involves several latent and gradient computations, there is no significant difference in peak memory compared to R2F since it follows a sequential process. R2F requires approximately 31GB of peak memory, and R2F+ requires approximately 35GB of peak memory where an additional 4GB mostly comes from the gradient computations in cross-attention control.
>
> **GPU Time.** The time taken for R2F+ is approximately 72 sec, which can be decomposed as 1) masked latent generation via object-wise R2F takes around 42 sec, and 2) region-controlled concept guidance takes around 30 sec.  Specifically, for the process of 1), the generation of each object takes around 20 sec (same as R2F) with an additional 1 sec for masking, resulting in a total of 42 sec for two objects. For the process of 2), the majority of the increased computation time compared to R2F is attributed to attention control, which adds around 10 sec. Consequently, for a prompt with N objects, the time complexity of R2F+ is expected as N*(T+1)+(T+10), where T is the inference time of R2F or SD3.
>
> Thus, in our experiments, we generated each image **within tens of seconds to a few minutes on a single 40GB A100 GPU, which is feasible.**
>
> - **We added this efficiency study in Section O of the revised manuscript.**

---

> > ### Author Response · Authors · 2024-11-19
> > **Author's Response to Reviewer zqku (2/3)**
> >
> > `Q1. I’m not entirely clear on the specific rules LLMs use to determine the “visual detail level.” In your writing, this measure is used in alternating concept guidance to set the guidance length for rare and frequent prompts, with more challenging rare concepts requiring extended guidance. However, LLMs lack knowledge of diffusion priors, which would inform the difficulty associated with generating certain objects or attributes.`
> >
> >
> > It is known that the diffusion denoising process determines rough global features (e.g., shape) in the initial steps and decides detailed local features (e.g., texture) in the later steps. Therefore, we used **the degree of locality required to draw each visual concept** as a specific rule for determining the visual detail level. As shown in Table 9 of Appendix B, this rule is carefully reflected in the full LLM instruction as follows:
> >
> > > d. Additionally, please provide how much local visual detail is needed to draw the rare concept on a scale of 1 (minimal detail needed) to 5(local detail essential), and explain why. Please give the score according to the degree of locality used to draw the visual concept.
> >
> > In addition, we further provide the LLM with in-context examples for each visual detail level from 1 to 5, enabling more precise scoring (See Table 10).
> >
> >
> > As a result, R2F with a visual detail level-aware stop point consistently outperformed R2F with a fixed stop point across various concept categories (See Figure 9). This demonstrates that the **zero-shot ability of the LLM** to assess the visual detail level of a concept enables more accurate concept guidance even **without the need for diffusion priors**. Nevertheless, as you suggested, reflecting diffusion priors more accurately to the LLM could further enhance compositional generation performance, and we would like to leave this as future work.
> >
> >
> > `Q2. The example you give in Figure 4, where "plants made of glass", I don't think it is a frequent concept. Furthermore, in the initial stages of denoising, diffusion models primarily focus on generating rough visual features (e.g., shape, location). Consider the concept of “furry”; both “furry bird” and “furry tiger” are frequent concepts LLMs may output, yet there is a significant difference in the size and shape of these objects, which has a notable impact on the generated result. Thus, I question whether LLMs can reliably provide suitable frequent concepts.`
> >
> > Thanks for your careful review. The goal of R2F is not to obtain concepts that are absolutely frequent, but rather to obtain **relatively** frequent concepts that can yield benefits in concept composition. From this perspective, "plants made of glass" is relatively more frequent than "cactuses made of glass", which can lead to performance improvements. Also, in the full LLM instruction (See Table 9), we prompted LLM to identify frequent concepts that **should be relevant** to the original rare concept as follows:
> > > ...when a rare concept is identified in the input text, you **should** replace it with **relevant yet more frequent** concepts.
> >
> > With this careful instruction, LLMs can reliably provide suitable frequent concepts. Usually, the generated frequent concepts often contain general terms easier for composition such as "animal" or "object" (See Table 14 of Appendix H for more detailed examples of the generated frequent concepts). As a result, unreliable mappings, such as the substitution of "bird" or "tiger" when the prompt context is unrelated, are unlikely to occur.
> >
> > Therefore, we believe that our approach can **reliably obtain suitable frequent concepts from LLMs with careful instructions**.

---

> ### Author Response · Authors · 2024-11-19
> **Author's Response to Reviewer zqku (3/3)**
>
> `Q3. Is the design of R2F+ necessary？ In fact, layout-based methods have outstanding spatial awareness, however, the trade-off is increased computational cost and a decline in image quality (in terms of detail, aesthetics, etc.). First, you need to conduct a comparative evaluation of R2F+ in terms of image quality. Additionally, as noted in Table 3’s T2I-CompBench, R2F achieves higher spatial metrics than both the layout-based method LMD and the LLM-based method RPG. Thus, expanding R2F to a layout-based approach may be unnecessary, as it would only improve spatial performance while significantly compromising image quality.`
>
> **Image Quality.** Per your suggestion, we additionally measured three popular image quality scores (e.g., LAION-aesthetic [5], PickScore [6], and ImageReward [7]) of R2F+ for multi-object cases in RareBench and compared it to R2F. As shown in the table below, there was no significant difference in image quality scores.
>
> | Image Quality Scores | LAION-aesthetic | PickScore | ImageReward |
> | -------- | -------------- | -------------- | -------------- |
> | R2F      | 3.980 +- 0.361 | 0.226 +- 0.009 | 0.626 +- 0.029 |
> | R2F+     | 3.887 +- 0.353 | 0.222 +- 0.009 | 0.609 +- 0.033 |
>
> - **We added this quality analysis in Section N of the revised manuscript.**
>
> **Necessity of R2F+.** R2F+ is useful in many applications where the layout-aware composition is very important. **(1) Layout-aware image/poster design.** When a user creator wants to place an object in a specific position within an image for poster design, R2F is insufficient because it cannot adjust absolute positions. In such cases, R2F+ becomes essential. **(2) Data synthesis for enhancing spatial understanding of foundation models.** Recent multi-modal LLMs (e.g., LLaVA [8]) and pre-trained VLMs (e.g., CLIP [9]) are known to exhibit weaknesses in spatial understanding [10], so several studies have attempted to enhance their performance with spatiality-aware image synthesis (e.g., generating images that accurately captures spatial information in text prompts). R2F+ has the potential to enhance the performance of these foundation models by serving as a data synthesis method, as spatial composition is more critical than image quality in this case.
>
> - **We added this discussion in Section P of the revised manuscript.**
>
> ---
> [5] Laion Aesthetic Predictor. https://github.com/LAION-AI/aesthetic-predictor, 2022
>
> [6] Pick-a-Pic: An Open Dataset of User Preferences for Text-to-Image Generation, NeurIPS, 2023
>
> [7] Imagereward: Learning and Evaluating Human Preferences for Text-to-image Generation, NeurIPS, 2024
>
> [8] Visual Instruction Tuning, NeurIPS, 2024
>
> [9] Learning Transferable Visual Models From Natural Language Supervision, ICML, 2021
>
> [10] SpatialVLM: Endowing Vision-Language Models with Spatial Reasoning Capabilities, CVPR, 2024
>
>
> `Q4. You can consider using the IterComp[1], which is a backbone specifically designed for compositional generation and may lead to a more significant performance improvement.`
>
> Thanks for introducing an important relevant work. We conducted additional experiments using IterComp as the backbone for R2F, and the results are in the below table.
>
>
> | Models | Property | Shape | Texture | Action | Complex | Concat | Relation | Complex |
> | ------------ | ---- | ---- | ---- | ---- | ---- | ---- | ---- | ---- |
> | SDXL         | 60.0 | 56.9 | 71.3 | 47.5 | 58.1 | 39.4 | 35.0 | 47.5 |
> | R2F_sdxl     | **71.3** | **71.9** | **73.8** | **54.4** | **70.6** | **50.6** | **36.0** | **52.8** |
> | **IterComp**     | 63.8 | 66.9 | 61.3 | 65.6 | 61.9 | 41.3 | 29.4 | 53.1 |
> | **R2F_itercomp** | **78.1** | **77.5** | **79.4** | **66.9** | **63.9** | **41.5** | **36.6** | **53.4** |
> | SD3.0        | 49.4 | 76.3 | 53.1 | 71.9 | 65.0 | 55.0 | 51.2 | 70.0 |
> | R2F_sd3.0    | **89.4** | **79.4** | **81.9** | **80.0** | **72.5** | **70.0** | **58.8** | **73.8** |
>
> Overall, R2F_itercomp **consistently improves** the compositional generation performance of IterComp on Rarebench dataset (i.e., better T2I alignment scores by GPT-4o). These results further demonstrate the flexibility of R2F across the diffusion backbones. R2F_itercomp was generally better than R2F_sdxl but worse than R2F_sd3.0. This is likely because IterComp enhances the SDXL backbone by compositional-aware preference optimization [11], so it might not yet match the generative performance of the more recent SD3.0 backbone. We will include these results in Table 4 of the main paper.
>
> - **We included this result in Table 4 of the revised manuscript.**
>
> ---
> [11] IterComp: Iterative Composition-Aware Feedback Learning from Model Gallery for Text-to-Image Generation, ArXiv, 2024

---

> > ### Author Response · Authors · 2024-11-24
> > **Follow-up**
> >
> > Dear Reviewer,
> >
> > We would appreciate your letting us know if our response has addressed your concerns.
> > Thank you for your time and effort in the rebuttal period.
> >
> > Best regards

---

> ### Comment · Reviewer_zqku · 2024-11-24
> **Official Comment by Reviewer zqku**
>
> Thank you for your detailed response. After reading all the replies, I plan to keep my score for acceptance. I appreciate this thoughtful  discussion.

---

### Official Review · Reviewer_zTAb · 2024-10-31

**Soundness:** 4
**Presentation:** 4
**Contribution:** 4
**Rating:** 8
**Confidence:** 4

**Summary:**

This paper examines diffusion-based image generation for objects with unusual attributes, which is termed as rare composition of concepts and pretty common in art design. Current methods are struggle to accurately generate images from rare and complex prompts. To solve this question, this approach effectively utilizes the correlation between frequent and common composition. Specifically, in the early stage of the reverse process, the frequent composition is used to guide noise prediction where the rare one is used. In this way, the frequent one is used to provide good initialization for the final generation. This method is training free with both theoretical analysis and experimental validation provided. An advanced version of the region-based generator is also proposed.

**Strengths:**

1.	The observation in alternating prompts in diffusion-based models are important.
2.	Both global and region-based generation are proposed.
3.	Detailed visualization are provided.

**Weaknesses:**

1.	The current design for the scheduling of the selection of frequent and rare composition of concepts is a bit ad-hoc. You always use frequent composition at the begining and then start randomly selection of composition after a fixed point. Based on your theoretical analysis, any additional guidance can be included or used to determine the selection of composition of concepts?
2.	From your example, each rare composition has only two concept. How do you generalize your approach to more complicated and rare composition (3 or more concepts, such as adj. + adj. + noun, e.g., an agent rabbit with a gun in a casual suit ).
3.	Have you tried to use rare components at the beginning and then use frequent instead? The intuitive explanation for using frequent one first is needed.  It will be good if you have relevant experimental results.
4.	In real world, both rare and frequent composition of concepts are considered in generation. Then, the method that improves quality of rare composition should not hurt the quality of frequent composition. Without manual determination, how can your approach still maintain high generation of frequent composition. In other words, it will be good if you can provide discussion on how your method could be adapted to automatically handle both rare and frequent compositions without manual intervention.

**Questions:**

Please address my questions in the weakness.

---

> ### Author Response · Authors · 2024-11-19
> **Author's Response to Reviewer zTAb**
>
> We sincerely appreciate the reviewers' constructive comments and positive feedback on our manuscript.
>
> `W1. The current design for the scheduling of the selection of frequent and rare compositions of concepts is a bit ad-hoc. You always use frequent composition at the beginning and then start randomly selecting of composition after a fixed point. Based on your theoretical analysis, any additional guidance can be included or used to determine the selection of composition of concepts?`
>
>
> We thank the reviewer for your constructive comments. We acknowledge your concerns but respectfully argue that our scheduling approach is not ad-hoc but carefully designed for two reasons.
> **(1) Leveraging abundant knowledge in LLMs**. To ensure *careful* scheduling, we leverage the strong zero-shot ability of LLMs to extract "visual detail level" required to draw each concept and use it for concept guidance, based on prior observations that diffusion models determine rough visual features during the early sampling steps and detailed visual features in the later steps (as explained in lines 237-240). This LLM-guided scheduling approach provides an appropriate concept guidance schedule across prompts with various semantics.
> **(2) Consistent performance improvement across diverse concept categories.** With our careful scheduling, R2F significantly improves the composition performance across diverse concept categories, including property, shape, texture, etc, outperforming fixed scheduling as shown in Figure 9.
>
>
> `W2. From your example, each rare composition has only two concepts. How do you generalize your approach to more complicated and rare composition (3 or more concepts, such as adj. + adj. + noun, e.g., an agent rabbit with a gun in a casual suit )`
>
>
> Thanks for your careful comments. RareBench already includes the complicated rare composition cases (as the '*complex*' case), consisting of three or more concepts, and R2F still exhibits superior performance on such complex cases as shown in Table 6. Specifically, looking at Figure 6, there is an example "A horned bearded spotted raccoon smiling" from the complex case, and R2F successfully generates the image that accurately follows the prompt. Technically, given examples such as "adj1 + adj2 + noun", R2F finds a noun that more frequently appears in the context of "adj1 + adj2", and uses it for frequent concept guidance.
>
>
> `W3. Have you tried to use rare components at the beginning and then use frequent instead? The intuitive explanation for using frequent one first is needed. It will be good if you have relevant experimental results.`
>
>
> As per your suggestion, we generated images using rare concepts at the beginning and then used frequent concepts at the last steps instead. The generated images can be found in Figure 17 of Appendix L. Overall, the generated image tends to align more closely with the frequent prompt rather than the original rare prompt. This is because diffusion is a step-wise denoising process where the generated image depends more on the lately used prompt. Therefore, it is advisable to guide the process by using frequent concepts first and rare concepts last.
>
> - **We included these results in Section L of the revised manuscript.**
>
> `W4. In the real world, both rare and frequent composition of concepts are considered in generation. Then, the method that improves the quality of rare compositions should not hurt the quality of frequent composition. Without manual determination, how can your approach still maintain a high generation of frequent composition. In other words, it will be good if you can provide discussion on how your method could be adapted to automatically handle both rare and frequent compositions without manual intervention.`
>
>
> Thanks for your constructive comments. Our R2F framework **leverages the LLM** to determine whether each decomposed sub-prompt has a rare concept and needs frequent concept guidance **without requiring manual intervention**. For example, as shown in Figure 4, for the sub-prompt "an awful snake" (denoted as $c^2$), the LLM determines it has no rare concepts, and thus, R2F guidance is not applied for this sub-prompt. As a result, on the **T2I-compbench**, containing a high proportion of frequent concepts, R2F **still shows superior composition performance** indicating that it can effectively control the quality of frequent concept composition. Furthermore, with the region-guided R2F+, we can ensure that addressing rare compositions does not compromise the quality of frequent compositions.

---

> > ### Comment · Reviewer_zTAb · 2024-11-24
> > **Thank you for your effort and feedback**
> >
> > Thanks for your effort in the rebuttal and clarification. I will keep my score.

---

### Official Review · Reviewer_p5vc · 2024-11-03

**Soundness:** 3
**Presentation:** 3
**Contribution:** 3
**Rating:** 8
**Confidence:** 4

**Summary:**

This paper studies how to perform rare concept image generation with current pre-trained diffusion models. The authors leverage the LLMs to extract the rare concept and rewrite the prompt, and then perform rare-to-frequent guidance with the rewritten prompts across the multi-step denoising generating process. Abudent theoretical and empirical analyses are provided to validate the effectiveness of the method.

**Strengths:**

1) The proposed rare-to-frequent prompt rewrite is novel and effective in terms of generating rare-concept-images.
2) The empirical results looks promising.
3) Solid empirical results are provided to validate the effectiveness of the method.
4) A new benchmark, RareBench, is provided to facilitate research in the task of rare-concept-image-generation.
5) Code and detailed implementation is provided to ensure the reproducibility of the method.

**Weaknesses:**

(1) The method requires alternating among a set of prompts during denoising process, which makes multiple step inference inevitable. Therefore, this design might not work well with current state-of-the-art acceleration methods, which reduce the number of denoising steps to 4 steps or even less.

(2) There is a small gap between the theoretical analysis and the empirical method. For the theoretical analysis, the author study the scenarios of linearly interpolation of scores produced by different prompts. While for empirical results, the author performs alternating prompts across different denoising steps.

**Questions:**

Please see weakness (1). The reviewer is curious about how can we apply the proposed method on accelerated version of diffusion models such as consistency model.

---

> ### Author Response · Authors · 2024-11-19
> **Author's Response to Reviewer p5vc**
>
> `W1. The method requires alternating among a set of prompts during the denoising process, which makes multiple-step inference inevitable. Therefore, this design might not work well with current state-of-the-art acceleration methods, which reduce the number of denoising steps to 4 steps or even less.`
>
>
> We thank the reviewer for helping us improve our paper. To address your concern, we conducted additional experiments using FLUX-schnell, one of the state-of-the-art diffusion models that can generate high quality images in 4 steps, as the backbone for R2F with 4 steps of denoising process.
>
> The core idea of R2F is that by **exposing frequent concepts** to the diffusion sampling process its rare concept composition performance can be significantly enhanced. Prompt alternating is one design choice, which is highly effective (in terms of T2I alignment) and efficient (as only one prompt, either rare or frequent, is used for each step) for long guidance length. However, as you pointed out, the efficacy of the prompt alternating may diminish for short guidance length. In this case, we recommend using the Composable approach (detailed in Section 4.4) which blends rare and frequent concepts within the text embedding as an alternative approach.
>
> **Configuration.** For the short guidance length of 4 steps, we adjusted the interpolation configuration of the Composable method. For concepts with a visual detail level from 1 to 3, we applied the composable method only for the first step, while for those with a visual detail level from 4 to 5, we applied it to both the first and second steps. In the final third and fourth steps, only the original rare prompt was exposed. We set the blending factor of $\alpha$ to 0.3.
>
> **Result.** The image generation results are presented in the table below, and the **generated images are added in Figure 16 of Appendix K in the revised manuscript**. Similar to the original results with a longer guidance length, the T2I alignment performance of FLUX-schnell improved in most cases when R2F was applied. Therefore, the frequent concept exposure idea of R2F can be generalized to the latest acceleration methods with 4 steps, which can have a broader impact on many real-world applications where fast inference time is crucial.
>
> | RareBench | Property | Shape | Texture | Action | Complex |
> | -------------------------------- | ---- | ---- | ---- | ---- | ---- |
> | FLUX.1.schnell (4 steps)         | 72.5 | 68.1 | 49.3 | 61.2 | 73.7 |
> | R2F_flux.1.schnell (4 steps)     | 78.7 | 75.0 | 56.8 | 67.5 | 68.7 |
>
>
> - **We included these results in Section K of the revised manuscript**.
>
>
>
> `W2. There is a small gap between the theoretical analysis and the empirical method. For the theoretical analysis, the author studies the scenarios of linear interpolation of scores produced by different prompts. For empirical results, the author performs alternating prompts across different denoising steps.`
>
>
> Thanks for your careful comments. Our theory assumes a score estimator for image sampling, which does not include multi-step denoising. For diffusion models with multi-step denoising, both the linear interpolation (of latents or text embeddings) and alternating prompts can be regarded as a way of interpolation. Table 6 empirically compares the effectiveness of these design choices, and the alternating approach is the most effective in multi-step denoising. Additionally, the alternating approach is the most efficient because it only requires to use either the rare or frequent prompt at each step. Nevertheless, both the linear interpolation and the alternating approach are more effective than the vanilla SD3.0.
>
> - **We clarified this explanation in lines 213-215 in Section 3.2 of the revised manuscript.**

---

> > ### Comment · Reviewer_p5vc · 2024-11-19
> >
> > Thanks for the added experiments and clarification, which have resolved my concerns. I have increased the score accordingly.

---

> > > ### Author Response · Authors · 2024-11-20
> > > **Thanks for your positive feedback**
> > >
> > > We are glad to hear that you are satisfied with our response. Again, thank you very much for your insightful comments.

---

### Author Response · Authors · 2024-11-19
**General Response**

We sincerely appreciate all the reviewers' positive feedback and valuable comments. Most reviewers agreed that (1) **the observation and methodology are novel** (All reviewers), (2) **empirical results are solid and promising** (All reviewers), (3) **the presentation is clear and reproducible** (Reviewer p5vc, zqku, and 7ZtE), and (4) **the proposed benchmark looks significant** (Reviewer p5vc and 7ZtE). During the rebuttal, we addressed the reviewers' remaining concerns by providing clarifications with additional experimental results (including two additional latest diffusion backbones, three image quality metrics, GPU efficiency analysis, and more visualizations; see the revised PDF file). We hope that the remaining concerns are successfully addressed by the rebuttal and are happy to answer more questions during the discussion period.

---

We thank all reviewers for their constructive comments and insightful suggestions. We have uploaded the final revised manuscript, which includes the following modifications and improvements:

Major (enhanced practical applicability)
- Integration with two more recent backbones: IterComp (in Section 4.3) and FLUX (in Appendix K)
- Faster inference; supporting 4-step inference with FLUX integration (in Appendix K)
- GPU time and memory analysis (in Appendix O)
- Discussion for applications (in Appendix P)

Besides
- Clarification of the connection between theory and method (in lines 213-215 of Section 3.2)
- Three image quality scores (in Appendix N)
- LLM prompting study using LAION dataset (in Appendix M)

Changes are highlighted in blue.
Thanks again for the constructive efforts in the comments and reviews.

Authors

---

### Meta-Review · Area_Chair_WDCX · 2024-12-19

**Metareview:**

This paper proposes an interesting approach for compositional image generation with rare concepts. Given the text prompt with rare concepts, the LLM first decomposes the prompt into regions and then maps rare concepts into frequent ones, which further guides the diffusion sampling process. This paper receives four positive reviews. Reviewers clearly acknowledge the significant contribution, the interesting and inspiring observations, the novel idea, and the solid experiments. Reviewers questions on design choices, experiments, and others are adequately addressed during the rebuttal. All reviewers responded to the rebuttal and either increased the scores or kept the original positive scores. Therefore, AC would recommend acceptance (spotlight).

**Additional Comments On Reviewer Discussion:**

Reviewers raised questions on design choices, experiments, etc. Most of them are adequately addressed in the rebuttal, and all reviewers responded that they are satisfied with the rebuttal.

---

### Decision · Program_Chairs · 2025-01-22

Accept (Spotlight)